# Prompt as Knowledge Bank: Boost Vision-language model via Structural Representation for Zero-shot Medical Detection

**Yuguang Yang**[*,1,2], **Tongfei Chen**[*,3], **Haoyu Huang**[3,5], **Linlin Yang**[†,4], **Chunyu Xie**[†,2], **Dawei Leng**[‡,2], **Xianbin Cao**[1], **Baochang Zhang**[3,6]

[1]School of Electronic Information Engineering, Beihang University, China
[2]360 AI Research, Qihoo 360, China
[3]School of Artificial Intelligence, Beihang University, China
[4]State Key Laboratory of Media Convergence and Communication,
  Communication University of China, China
[5]National Superior College for Engineers, Beihang University, China
[6]Artificial Intelligence Research Center, Lobachevsky State University,
  Nizhny Novgorod 603022, Russia

## Abstract

Zero-shot medical detection enhances existing models without relying on annotated medical images, offering significant clinical value. By using grounded vision-language models (GLIP) with detailed disease descriptions as prompts, doctors can flexibly incorporate new disease characteristics to improve detection performance. However, current methods often oversimplify prompts as mere equivalents to disease names and lacks the ability to incorporate visual cues, leading to coarse image-description alignment. To address this, we propose StructuralGLIP, a framework that encodes prompts into a latent knowledge bank, enabling more context-aware and fine-grained alignment. By selecting and matching the most relevant features from image representations and the knowledge bank at layers, StructuralGLIP captures nuanced relationships between image patches and target descriptions. This approach also supports category-level prompts, which can remain fixed across all instances of the same category and provide more comprehensive information compared to instance-level prompts. Our experiments show that StructuralGLIP outperforms previous methods across various zero-shot and fine-tuned medical detection benchmarks. The code will be available at `https://github.com/CapricornGuang/StructuralGLIP`.

## 1 Introduction

Zero-shot medical detection is crucial in healthcare as it enhances detection capabilities without requiring additional annotated medical images, even after model fine-tuning (Badawi et al., 2024; Mahapatra et al., 2021; Qin et al., 2022). This is particularly valuable in clinical settings, where doctors often encounter new disease characteristics not previously documented. In such cases, clinicians can temporarily create custom prompts to guide the detection process, allowing models to adapt to novel scenarios more effectively. Recent studies have explored the potential of grounded language-image pre-training models (GLIP) (Phan et al., 2024; Tiu et al., 2022; Li et al., 2022c; Yao et al., 2022) to reduce dependence on annotations by leveraging prior knowledge. These models conduct detection by contrasting image features with descriptive texts, known as *contextual prompts*, generated by visual question-answer models for query objects. To adapt GLIP to the medical domain, recent works (Qin et al., 2022; Wu et al., 2023b; Guo et al., 2023) have employed medically enhanced question-answer models like PubMedBERT (Gu et al., 2021) and BLIP (Li et al., 2022a) to create

---

*: Co-First Authors. {guangbuaa, tfchen}@buaa.edu.cn
†: Corresponding Authors. lyang@cuc.edu.cn, xiechunyu@360.cn
‡: Project Lead.

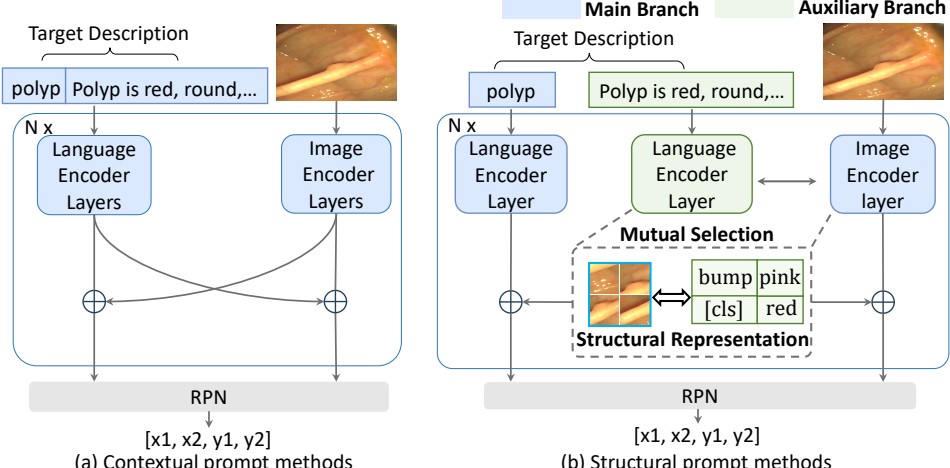

Figure 1: (a) Contextual prompt methods directly concatenate the prompt and target. (b) Our structural prompt method encodes prompts into a latent knowledge bank.

attribute-rich prompts. These prompts capture nuanced characteristics of query targets, improving domain adaptation and performance beyond traditional supervised training.

However, existing contextual prompt-based methods often suffer from coarse alignment between images and target descriptions, resulting in two key issues. **First**, these methods typically treat prompts as contexts that are equivalent to the target, easily causing distribution shift problems to the target's representation. Despite incorporating the prior about the target, they also introduce distracting information about the target. This leads to misalignment between the target and the actual visual cues in the image (see Fig. E). **Second**, category-level descriptions can not be effectively encoded within the context, which often contain ambiguous vocabularies such as "tissue with pink or red color, irregular or round shape" for a "bump". This causes that the most relevant prompt can not be precisely matched with the input image.

To address the aforementioned issues, we present StructuralGLIP, a novel zero-shot medical detection model that derives *structural representations*, which are delicately organized sets of features specifically designed to represent the nuances of the target and the input image. Specifically, as shown in Fig. 1, instead of simply concatenating prompts with the target, StructuralGLIP adopts a dual-branch architecture. The main branch processes the target name and input image, while the auxiliary branch encodes the prompts into a latent knowledge bank. At each layer, rather than directly performing cross-modal fusion between vision and language features, StructuralGLIP introduces a mutual selection mechanism. This mechanism matches vision features from the main branch with relevant prompt features stored in the latent knowledge bank, where we extract latent prompt tokens and latent vision tokens that both highly relevant to the target and the current input image, forming fine-grained structural representations. Once these structural representations are formed, the image and language features from the main branch are fused with the selected prompt tokens via cross-modality multi-head attention (Vaswani et al., 2017). This enhances the overall feature alignment and improves the fusion process within the main branch. Conceptually, the hierarchical knowledge bank in StructuralGLIP functions like a memory system (Bi, 2021; Paivio, 2013). As the image is processed, relevant knowledge is dynamically retrieved from the bank. This enables the model to better align the image features with the prompt information, resulting in more accurate and context-aware detection.

In this way, StructuralGLIP can address the challenge of effectively utilizing category-level prompts, which provide broader yet consistent information for all instances within the same category (see Fig. B for visualization). StructuralGLIP's instance-wise selection mechanism ensures that even fixed category-level prompts are dynamically aligned with the specific visual features of each instance. This not only improves detection precision but also enhances efficiency, as category-level prompts can remain fixed across instances of the same category. To validate the proposed method, we benchmark StructuralGLIP against previous state-of-the-art methods on eight datasets under endoscopy, microscopy, photography, and radiology four imaging conditions, and conduct a comprehensive

analysis towards StructuralGLIP's structural representations. The primary contributions of our work are as follows:

- We introduce StructuralGLIP, a novel architecture that achieves adaptive, context-aware alignment between visual features and target descriptions by utilizing a dual-branch structure with mutual selection, enhancing the precision of medical object detection.
- We propose the use of category-level prompts, which remain fixed for all instances of the same target. Unlike instance-level prompts, category-level prompts provide more comprehensive prior knowledge about the target disease, reducing the need for prompt generation for each individual image while maintaining strong detection performance.
- We explore zero-shot medical detection in more practical settings by demonstrating how zero-shot enhancement can further improve the performance of models fine-tuned on medical data. StructuralGLIP not only surpasses fully supervised methods such as RetinaNet but also seamlessly integrates into GLIP models fine-tuned on medical datasets, achieving an average improvement of +4% AP.

## 2 RELATED WORK

**Zero-shot medical detection** aims to identify and locate pathology concepts in medical images without relying on annotated data from the target domain (Vilouras et al., 2024; Qin et al., 2022; Paul et al., 2021; Mahapatra et al., 2021; Sahasrabudhe et al., 2020; Le Bescond et al., 2022). Classical strategies include cross-domain generalization (Bian et al., 2022; Bansal et al., 2024) and unsupervised learning (Sahasrabudhe et al., 2020; Le Bescond et al., 2022; Paul et al., 2021). Cross-domain generalization utilizes data from related domains under varied conditions, such as different imaging techniques (Bian et al., 2022) or demographic differences (Bansal et al., 2024), to adapt models across diverse scenarios. Unsupervised learning methods leverage side information to bypass direct supervision, such as using cell nuclei structure for image resolution analysis (Sahasrabudhe et al., 2020), employing GANs with public annotations to enhance mask quality (Le Bescond et al., 2022), and correlating medical reports with disease features to increase detection accuracy (Paul et al., 2021). However, these methods are often tightly coupled to specific data priors and exhibit a considerable performance gap compared to supervised models, limiting their clinical significance.

Recent approaches have integrated expert-level knowledge into vision-language models trained on natural images to facilitate domain transfer (Liu et al., 2023a; Lai et al., 2024; Tiu et al., 2022; Wu et al., 2023a; Zhang et al., 2023). However, most of these efforts focus on medical classification, while the more practical and complex task of medical detection remains underexplored. For example, (Qin et al., 2022) conducted a comprehensive study on medical detection using prompts generated by a medically-enhanced language model, PubMedBERT (Gu et al., 2021). Follow-up studies (Wu et al., 2023b; Lu et al., 2023; Phan et al., 2024) employed BLIP (Li et al., 2022a) to generate image-specific linguistic attributes, or used GPT (Achiam et al., 2023) to detail target concepts with nuanced descriptions. Recent work (Guo et al., 2023) further advanced this approach by introducing an ensemble strategy for fusing multiple prompts to improve detection accuracy. However, these methods require unique prompts for each instance, significantly reducing efficiency. Our method, StructuralGLIP, addresses these challenges by introducing a vision-language model that leverages a knowledge bank to store a wide range of prompts, enabling instance-dynamic prompt selection in the latent feature space.

**Knowledge-bank-based prompt method** is initially developed for continual learning, which utilizes a prompt pool designed to enhance cross-domain generalization (Wang et al., 2022b;a; Smith et al., 2023; Wang et al., 2023; Du et al., 2022). Previous works (Wang et al., 2022b;a) select top-$k$ prompts aligned with input image features, facilitating domain-specific modeling. Recent advances have evolved this strategy, replacing the top-$k$ prompt selection with a more flexible continuous prompt fusion strategy (Smith et al., 2023), exploring its potential for vision-language model (Wang et al., 2023), and expanding applications to open-vocabulary detection tasks (Du et al., 2022). However, these methods typically require an additional training phase and are restricted to prompt retrieval in the input layer. In contrast, StructuralGLIP explores a linguistically accessible avenue by directly utilizing the attributes predefined by the generative models and embeds these attribute prompts into a hierarchy knowledge bank situated within an auxiliary branch to achieve a layer-wise selection process.

## 3 METHODOLOGY

### 3.1 PRELIMINARIES

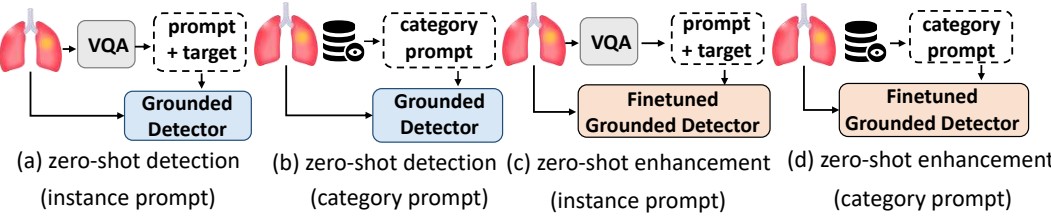

Figure 2: Experimental settings for zero-shot medical detection and enhancement.

**Zero-shot medical object detection** means improving the model's medical detection performance without the use of supervised image labels. This formulation emphasizes "further improvement without supervised images", which contains two experimental settings. Firstly, in the classical setting, the model, without fine-tuning on medical datasets, uses pre-trained vision-language models with prompts to infer medical concepts (see Fig. 2 (a) and (b)). Secondly, considering the clinical setting prefers supervised models for their excellent performance, we propose a zero-shot enhancement setting. This involves fine-tuning the model on medical datasets first, and then using prompts to further improve performance on unseen medical images, without requiring additional labels (see Fig. 2(c) and (d)). This setting mirrors real-world clinical needs, where models can be continuously improved with new knowledge without the need for labeled data.

**GLIP** redefines object detection as a phrase-grounding task by employing a late fusion dual-tower architecture to align image and text features. It uses separate backbones $Enc_I$ and $Enc_T$ to extract initial encodings $O^0$ and $P^0$ for images and text, respectively. These features are then integrated through a cross-modal multi-head attention module (X-MHA), enabling fine-grained interaction between the modalities. The integration of image and text features through the deep fusion module (X-MHA) is formalized as follows:

$$O^i_{t2v}, P^i_{v2t} = \text{X-MHA}(O^i, P^i), \tag{1}$$

$$O^{i+1} = f^i_I(O^i + O^i_{t2v}), \quad P^{i+1} = f^i_L(P^i + P^i_{v2t}), \tag{2}$$

where $f^i_I$ and $f^i_L$ are the $i^{\text{th}}$ encoder layers for images and text, respectively, and $i \in [1, N]$. After $N$ layers of interaction, the final image and text representations are denoted as $O^N$ and $P^N$, respectively. These representations are used as input to the RPN for generating object proposals:

$$R_{\text{GLIP}} = \text{RPN}(O^N, P^N), \tag{3}$$

where $R_{\text{GLIP}}$ denotes the set of region proposals of GLIP generated by the RPN. Each proposal $r \in R$ is characterized by its bounding box coordinates and a confidence score, indicating the likelihood of the region containing the target object.

### 3.2 ZERO-SHOT DUAL-BRANCH PROMPT FRAMEWORK

In the proposed StructuralGLIP framework, we introduce a novel zero-shot architecture to achieve fine-grained alignment between target description and medical images. The overall pipeline is shown in Fig. 3.

**Structurally separated auxiliary and main branches.** StructuralGLIP adopts a dual-branch architecture. The main branch processes the target name and input image, while the auxiliary branch encodes the prompts into a latent knowledge bank. Given the object target $T$ and the prompt $Prompt$, the initial representations $T^0$ and $B^0$ are obtained as follows:

$$T^0 = \text{Enc}_{L_1}(T), \quad B^0 = \text{Enc}_{L_2}(Prompt), \tag{4}$$

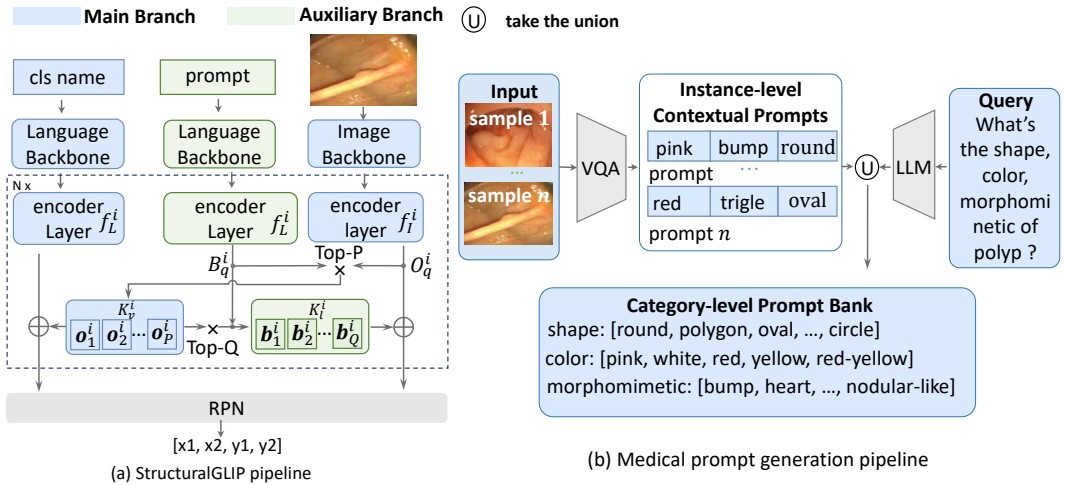

Figure 3: Pipeline of the proposed method and the automatic prompt generation.

where $\text{Enc}_{L_1}$ and $\text{Enc}_{L_2}$ are language backbones with shared parameters for the main and auxiliary branches, respectively. Here, $T^0$ and $B^0 \in \mathbb{R}^{N_l \times D}$ represent the initial encoded features of the target and prompts, with [PAD] tokens used to pad the input sentences to a uniform length $N_l$. The encoded prompts $B^0$ are then processed through the language encoder layers:

$$B^i = f^i_{L_2}(B^{i-1}), \tag{5}$$

where $f^i_{L_2}$ is the $i^{\text{th}}$ language encoder layer of the auxiliary branch, and $B^i$ denotes representation of prompt bank at the $i^{\text{th}}$ layer.

**Mutual prompt selection mechanism for structural representation in the auxiliary branch.** This mechanism identifies mutually relevant tokens between the visual tokens from the main branch and the linguistic tokens from the auxiliary branch. For selecting the Top-$P$ relevant visual tokens from the latent representation of the input image, we calculate their similarity with latent prompt features. The visual and linguistic representations in the $i$-th layer are denoted as $O^i_q \in \mathbb{R}^{N_v \times D}$ and $B^i_q \in \mathbb{R}^{N_l \times D}$, respectively. We have the following:

$$O^i_q = [\boldsymbol{o}^i_1, \boldsymbol{o}^i_2, \dots, \boldsymbol{o}^i_{N_v}], \quad \mathcal{K}^i_v = \text{Top-}P^{max}\left([\text{key} = \boldsymbol{o}^i_j, \text{value} = \boldsymbol{o}^i_j \boldsymbol{B}^i_q]^{N_v}_{j=1}\right), \tag{6}$$

where Top-$P^{max}$ ([key, value]) denotes selecting the keys with the Top-$P$ maximal values, $\mathcal{K}^i_v$ is the selected visual tokens in the $i$-th encoder layer, and $N_v$ is the token length of the visual encoder. Similarly, to select the Top-$Q$ tokens from the latent representation of the prompt, we use the similarity to the selected visual tokens $\mathcal{K}^i_v$:

$$B^i_q = [\boldsymbol{b}^i_1, \boldsymbol{b}^i_2, \dots, \boldsymbol{b}^i_{N_l}], \quad \mathcal{K}^i_l = \text{Top-}Q^{max}\left([\text{key} = \boldsymbol{b}^i_j, \text{value} = \boldsymbol{b}^i_j \mathcal{K}^i_v]^{N_l}_{j=1}\right), \tag{7}$$

where Top-$Q^{max}$ ([key, value]) denotes selecting the keys with the Top-$Q$ maximal values. $\mathcal{K}^i_l$ is the selected linguistic tokens in the $i$-th layer, and $N_l$ is the token length of the language encoder. These selected prompt tokens $\mathcal{K}^i_v$ and $\mathcal{K}^i_l$ are highly relevant to the target and the current input image, forming fine-grained structural representations.

**Deep fusion with the vision-language prompt in the main branch.** Once the structural representations are obtained, we serve these selected prompt tokens $\mathcal{K}^i_v$ and $\mathcal{K}^i_l$ as latent prompts for the deep fusion process of GLIP. Instead of using the auxiliary language encoder to enhance the features, the main branch's vision and language encoders leverage the knowledge from the selected tokens $\mathcal{K}^i_l$ and $\mathcal{K}^i_v$. This ensures that comprehensive knowledge from the prompts can be extracted precisely

and applied in an instance-wise manner to enhance the detection process. Specifically, we employ a multi-head attention (MHA) mechanism Vaswani et al. (2017) for $(\mathcal{K}_v^i, T_q^i)$ and $(\mathcal{K}_l^i, O_q^i)$:

$$O_{t2v}^{\mathrm{top}Q} = \mathrm{MHA}(\mathbf{Q} = \mathcal{K}_v^i, \mathbf{KV} = T_q^i) \quad T_{v2t}^{\mathrm{top}P} = \mathrm{MHA}(\mathbf{Q} = \mathcal{K}_l^i, \mathbf{KV} = O_q^i), \qquad (8)$$

where Q denotes the query item and KV denotes the key and value items for MHA, and $O_{t2v}^{\mathrm{top}Q}$, $T_{v2t}^{\mathrm{top}P}$ is the input image and target representations that incorporate the prior knowledge about the target from the selected tokens, respectively. These representations are then combined with the original layer representation using the following residual connection:

$$O^{i+1} = f_I^i(O^i + O_{t2v}^{\mathrm{top}Q}), \quad T^{i+1} = f_L^i(T^i + T_{v2t}^{\mathrm{top}P}). \qquad (9)$$

This deep fusion mechanism ensures that the model dynamically integrates relevant prompts at each layer, significantly enhancing instance-specific adaptation for zero-shot medical detection. After $N$ layers of interaction, we obtain the final image and text representations from the main branch, denoted as $O^N$ and $T^N$, respectively. Here, $T^N$ represents the target's representation, which has fused prompt information relevant to the current instance, achieving a more precise alignment with $O^N$. These representations are then used as input to the RPN for generating object proposals:

$$R_{\mathrm{StructuralGLIP}} = \mathrm{RPN}(O^N, T^N), \qquad (10)$$

where $R_{\mathrm{StructuralGLIP}}$ denotes the set of region proposals generated by the RPN in StructuralGLIP. This process effectively combines the structural representations from both the visual and language modalities to achieve accurate and context-aware zero-shot detection.

### 3.3 Instance/Category-level Medical Prompt Automatic Generation

As shown in Fig. 3(b), we propose a dual-level prompt generation mechanism that constructs a comprehensive prompt repository at both the instance and category levels. This enables StructuralGLIP to dynamically apply the most relevant knowledge during inference, significantly improving detection accuracy.

**Instance-level Prompt Generation.** For each medical image, we generate instance-specific prompts to capture unique visual features such as shape, color, and morphology using a Visual Question Answering (VQA) model like BLIP Li et al. (2022a). We query the model with targeted questions (e.g., "What is the shape of the polyp?"), and the responses form a set of instance-level contextual prompts (e.g., "[pink-white, bump-like, round]"). This process ensures that the model can dynamically adapt to the specific characteristics of each image, providing fine-grained descriptions that are crucial for precise detection.

**Category-level Prompt Generation.** In parallel, we construct a category-level prompt bank containing general attributes relevant to each medical category. Using a language model like GPT-4, we generate detailed descriptions for common attributes such as shape, color, and morphology (e.g., "typical shapes of polyps include round, oval, and nodular-like"). This enriched prompt bank serves as a static reference, enabling the model to capture the broader context of each category and generalize effectively across diverse medical cases. Finally, we gather all attributes from the instance-level prompt and concatenate them with the GPT-4 augmented results to derive the category-level prompt (displayed in Appendix G).

**Application.** Category-level prompts provide comprehensive information for entire classes of medical images and remain fixed across all images within the same category, offering higher efficiency compared to instance-specific prompts. Despite this advantage, prior methods Guo et al. (2023); Qin et al. (2022); Wu et al. (2023b) have not fully benefited from general prompts (see Tab. 2) due to their lack of adaptive prompt selection. StructuralGLIP, however, utilizes an instance-wise selection mechanism that supports category-level prompts effectively. This allows the model to dynamically select the most relevant prompts from the prompt bank, achieving performance comparable to or even better than instance-level prompts on certain datasets. This demonstrates that our method can efficiently leverage general prompts to enhance zero-shot detection without the need for instance-specific generation.

Table 1: Comparative experiment results on zero-shot medical detection across seven datasets, where **gray-shaded rows** represent the instance-level prompt results, while the unshaded rows represent the category-level prompt results.

| Methods | CVC-300 | | Kvasir | | ColonDB | | ClinicDB | | ETIS | | ISIC 2016 | | BCCD | | Avg. | |
|---|---|---|---|---|---|---|---|---|---|---|---|---|---|---|---|---|
| | AP | AP50 | AP | AP50 | AP | AP50 | AP | AP50 | AP | AP50 | AP | AP50 | AP | AP50 | AP | AP50 |
| GLIP | 29.8 | 37.9 | 25.9 | 33.6 | 21.7 | 32.4 | 22.1 | 29.6 | 6.7 | 9.7 | 10.5 | 20.0 | 8.9 | 18.4 | 17.9 | 25.9 |
| MIU-VL | 36.5 | 66.6 | 28.7 | 36.6 | 19.8 | 35.6 | 28.2 | **40.6** | 9.4 | 15.4 | 21.7 | 35.7 | 11.4 | 20.4 | 22.2 | 35.8 |
| AutoPrompter | 52.7 | 70.6 | 30.4 | 39.7 | 31.9 | 45.9 | 22.0 | 30.6 | 17.7 | 26.5 | 19.9 | 32.9 | 12.9 | 22.3 | 26.7 | 38.3 |
| Ours (instance) | **54.3** | **72.8** | **34.7** | **43.1** | **35.3** | **51.3** | **28.6** | 38.2 | **22.2** | **31.9** | **27.7** | **40.8** | **13.5** | **24.1** | **30.9** | **43.1** |
| MPT *w.* WBF | 3.27 | 9.40 | 12.2 | 14.4 | 14.2 | 19.1 | 11.2 | 14.0 | 12.0 | 17.0 | 1.13 | 5.37 | 1.22 | 4.75 | 7.8 | 12.0 |
| MPT *w.* Cluster | 36.7 | 47.5 | 12.0 | 17.0 | 11.9 | 21.4 | 11.2 | 14.0 | 12.0 | 17.0 | 19.8 | 30.9 | 14.3 | 33.8 | 16.8 | 25.9 |
| Ours (category) | **63.9** | **89.8** | **42.0** | **50.5** | **42.1** | **66.0** | **42.0** | **57.0** | **30.4** | **40.3** | 21.8 | 33.5 | **23.6** | **40.9** | **37.9** | **54.0** |

## 4 EXPERIMENT

We illustrate our experiment settings in Fig. 2, where we design four distinct settings to evaluate the model's performance. In Sec. 4.2, we follow traditional zero-shot setups to evaluate StructuralGLIP in a zero-shot setting without any fine-tuning on medical datasets, using both instance-specific and category-specific prompts (see Fig. 2(a) and (b)). In Sec. 4.3, we simulate clinical environments where supervised models are typically preferred. Here, we fine-tune the backbone of the proposed methods, *i.e.*, GLIP, (without using prompts) on medical datasets to form a refined detector. After fine-tuning, we incorporate linguistic prompts for the target disease into StructuralGLIP to perform zero-shot enhancement, evaluating the model's ability to improve performance even after fine-tuning (see Fig. 2(c) and (d)). The fine-tuned details are provided in Appendix D.

### 4.1 EXPERIMENTAL SETUP

**Datasets.** We select four types of medical imaging datasets involving eight benchmarks: 1) Endoscopy datasets for polyp detection: ClinicDB Bernal et al. (2015); Fernández-Esparrach et al. (2016), ColonDB Bernal et al. (2012), Kvasir Jha et al. (2020), ETIS Silva et al. (2014); 2) Microscopy dataset: BCCD shenggan et al. (2018) for blood cells detection; 3) Photography dataset: ISIC-2016 for skin lesions detection (benign lesion; malignant lesion); 4) Radiology image datasets: TBX11K Liu et al. (2023b) for tuberculosis detection in lung X-rays. Detailed elaboration is given in the Appendix C.

**Metric and baseline.** To evaluate our approach, we primarily benchmark against recent studies, mainly following Qin *et al.* Qin et al. (2022) (2023) and Wu *et al.* Wu et al. (2023b) (2023). Our baselines include recent GLIP-based methods (vanilla GLIP Li et al. (2022b), MIU-VL Qin et al. (2022), and AutoPrompter Wu et al. (2023b)) for instance-specific prompt generation setting, and works attempt to use category-specific prompt for detection (MPT Guo et al. (2023), and its variants MPT+SoftNMS Bodla et al. (2017), MPT+WBF Solovyev et al. (2021)). For zero-shot enhancement experiments, fully supervised detection models (RetinaNet Lin et al. (2020) and DyHead Dai et al. (2021)) are also included to provide a comprehensive evaluation landscape for our zero-shot enhancement experiments. The training details of the GLIP are elaborated in Appendix D.

### 4.2 RESULTS OF ZERO-SHOT MEDICAL DETECTION

**Superior transfer performance across various medical scenarios.** *For fairness, we ensured that all methods used consistent prompts for a fair comparison.* For instance-specific prompt methods, we utilized BLIP (Li et al., 2022a) as the vision-question answering model for all approaches, except for vanilla GLIP (Li et al., 2022b), which directly used the target name as text input. For MIU-VL (Qin et al., 2022), we additionally used PubMedBert Gu et al. (2021) to generate prompts specific to the target disease. AutoPrompter (Wu et al., 2023b) uses GLIP to produce the initial bounding box with instance-specific prompt and refine them with a self-training process with Yolo-X (Zheng et al., 2021). The experimental results are shown in Tab. 1 (instance-specific prompt), where all prompt methods enhance the original GLIP model's performance by providing additional descriptions. Among them, StructuralGLIP achieved the greatest improvement, with an average +4.2% AP ↑, +4.8% AP50 ↑

Table 2: AP% of vanilla GLIP and the proposed methods with instance-specific (I) and category-specific (C) prompt under zero-shot detection setting.

| | CVC-300 | | ClinicDB | | Kvasir | | Avg. | |
|---|---|---|---|---|---|---|---|---|
| | C | I | C | I | C | I | C | I |
| GLIP | 34.3 | 29.8 | 17.9 | 22.1 | 22.3 | 25.9 | 24.8 | 25.9 |
| ours | **63.9** | **54.3** | **42.0** | **28.6** | **42.0** | **34.9** | **48.3** | **39.2** |

Table 3: AP% of fine-tuned GLIP and the proposed methods with instance-specific (I) and category-specific (C) prompt under zero-shot enhancement setting.

| | CVC-300 | | ClinicDB | | Kvasir | | Avg. | |
|---|---|---|---|---|---|---|---|---|
| | C | I | C | I | C | I | C | I |
| GLIP | 70.0 | 67.5 | 54.3 | 63.0 | 44.5 | 51.1 | 56.2 | 60.5 |
| ours | **77.2** | **74.9** | **70.4** | **68.4** | **71.3** | **69.6** | **72.9** | **70.9** |

across seven datasets. We do not exhibit the results for the radiology dataset TBX-11k here, as the initial performance of the GLIP model on this dataset was poor, and the performance improvement for each prompt method is not distinguishable.

**Knowledge bank facilitates category-level prompts.** In this experiment, we focus on the effectiveness of category-level prompts generated by BLIP and GPT-4, which expand attributes related to the target across different dimensions such as colors, shapes, textures, and locations. These category-level prompts, being about 10 times longer than instance-specific prompts, remain consistent across all instances within the same class, and their details are provided in Appendix G. To benchmark against other methods, we include the MPT (Guo et al., 2023) approach, which is built upon the GLIP backbone and designed specifically to handle category-level prompts. MPT employs different prompt ensemble strategies, such as Weighted Box Fusion (WBF) and clustering, to split the category prompts into multiple groups and fuse the outputs for improved performance. Tab. 1 shows the performance comparison under different ensemble strategies. As seen in the results, StructuralGLIP achieves superior average performance across seven datasets compared to MPT. More importantly, StructuralGLIP consistently outperforms instance-specific prompt methods when utilizing category-level prompts (a +7% average AP ↑ across seven datasets). This suggests that StructuralGLIP can effectively harness the richer and more comprehensive information encoded in the category prompts.

We attribute this advantage to the dual-branch architecture of StructuralGLIP, where the prompts and image features are separated into an auxiliary and main branch, respectively. By introducing an instance-wise selection mechanism, StructuralGLIP can dynamically select the most relevant parts of the category prompt based on the input image. To further verify this, we directly feed the category prompt for GLIP to obtain GLIP's performance under the category prompt and follow MIU-VL to obtain its performance under the instance prompt, As shown in Tab. 2. The results demonstrate a significant improvement (average AP of $24.8 \rightarrow 49.3$) in StructuralGLIP's performance compared to the vanilla GLIP with category-level prompts. Interestingly, by comparing the performance of GLIP between using category-level prompt (see Tab. 2) and instance-level prompt (see Tab. 1), vanilla GLIP exhibit performance degradation when category prompts are employed (average AP of $25.7 \rightarrow 24.8$). In contrast, StructuralGLIP shows a significant AP improvement ($39.2 \rightarrow 49.3$). This highlights the advantage of StructuralGLIP's knowledge modeling and its ability to dynamically extract the most relevant prompt information for each instance, effectively leveraging the comprehensive knowledge provided by category-level prompts.

## 4.3 RESULTS OF ZERO-SHOT ENHANCEMENT FOR MEDICAL DETECTION

**StructuralGLIP surpasses the fully-supervised methods.** In this experiment, we evaluate zero-shot enhancement and also compare fine-tuned GLIP-based models with classic object detection models, such as FasterRCNN Ren et al. (2015) and RetinaNet Lin et al. (2020), which were fully supervised. As shown in Tab. 4, while the refined GLIP performs similarly to the supervised RetinaNet (55.2 *vs.* 56.6 average AP), incorporating instance-level prompts with StructuralGLIP raises the performance to 59.3 AP, a noTab. +2.7% improvement. For category-level prompts, StructuralGLIP achieves an average AP of 60.6, showing a slight improvement over instance-level prompts. However, *given that category-level prompts remain fixed across all images of the same class and can be pre-encoded in our auxiliary branch*, this performance boost comes with only the inference cost for calculating the attention matrix of prompt, further demonstrating the efficiency of our approach.

Table 4: Comparative zero-shot enhancement experiment results across datasets, s, where **gray-shaded rows** represent the instance-level prompt results, while the last unshaded block represents the category-level prompt results.

| Methods | Kvasir | | ColonDB | | ClinicDB | | ETIS | | CVC-300 | | ISIC 2016 | | BCCD | | TBX-11k | |
|---|---|---|---|---|---|---|---|---|---|---|---|---|---|---|---|---|
| | AP | AP50 | AP | AP50 | AP | AP50 | AP | AP50 | AP | AP50 | AP | AP50 | AP | AP50 | AP | AP50 |
| FasterRCNN | 63.4 | - | 44.1 | - | 71.6 | - | 44.5 | - | 59.4 | - | 50.3 | - | 56.9 | - | 33.9 | 73.9 |
| RetinaNet | 64.1 | - | 49.8 | - | 71.9 | - | 46.6 | - | 61.6 | - | **54.0** | - | 56.7 | - | 37.0 | 77.9 |
| GLIP | 64.8 | 82.2 | 56.8 | 79.1 | 65.1 | 82.6 | 60.4 | 77.0 | 75.2 | 95.9 | 39.9 | 50.9 | 55.4 | 78.2 | 35.2 | 75.3 |
| MIU-VL | 67.7 | 86.2 | 48.8 | 75.2 | 63.0 | 82.6 | 48.9 | 68.8 | 67.5 | **97.2** | 29.7 | 38.7 | 44.5 | 58.9 | 35.5 | 76.7 |
| AutoPrompter | **70.0** | 87.5 | 57.8 | **81.3** | 67.5 | 85.3 | 59.6 | 76.8 | **75.2** | 97.1 | 37.3 | 49.0 | 23.4 | 33.2 | 35.7 | 76.5 |
| Ours (instance) | 69.6 | **87.9** | 58.1 | 81.0 | **68.4** | **87.5** | 60.3 | 77.0 | 74.9 | 96.3 | 49.5 | **62.7** | 56.9 | 80.2 | 37.3 | 78.2 |
| MPT+Cluster | 25.1 | 30.0 | 22.3 | 29.5 | 24.8 | 29.3 | 24.7 | 29.8 | 33.4 | 41.5 | 25.6 | 33.7 | 22.8 | 30.6 | 31.4 | 68.2 |
| Ours (category) | **71.3** | **89.0** | **62.0** | **85.3** | **70.4** | **88.2** | **62.4** | **79.5** | **77.2** | 96.5 | 45.9 | **58.3** | **57.8** | **82.4** | **37.8** | **79.2** |

Table 5: Ablation results for Top-Q (y-axis) and Top-P (x-axis) on CVC-300 dataset with zero-shot medical detection seting.

| | 1 | 5 | 10 | 15 | 20 | 30 | 40 |
|---|---|---|---|---|---|---|---|
| 5 | 15.1 | 50.1 | 53.4 | 52.4 | 52.4 | 51.9 | 52.4 |
| 10 | 19.9 | 47.1 | 56.5 | 55.9 | 55.3 | 55.5 | 55.8 |
| 15 | 19.9 | 47.1 | 56.0 | 55.4 | 54.9 | 55.2 | 55.3 |
| 20 | 17.3 | 48.2 | 55.2 | 54.9 | 54.9 | 54.9 | 55.2 |
| 30 | 18.8 | 50.4 | 56.4 | 55.6 | 55.9 | 56.1 | 56.1 |
| 40 | 17.9 | 50.2 | 56.1 | 55.1 | 54.9 | 55.8 | 56.2 |

Table 6: Ablation study results (AP%) on the generation of category prompt using VQA and GPT across four datasets.

| Methods | Kvasir | ColonDB | ClinicDB | ETIS |
|---|---|---|---|---|
| MPT+VQA+GPT | 12.2 | 14.2 | 11.2 | 12.0 |
| Ours+VQA | 37.6 | 38.9 | 38.8 | 26.3 |
| Ours+VQA+GPT | **42.0** | **42.1** | **42.0** | **30.4** |

**StructuralGLIP facilitates further improvement on fine-tuned models.** Interestingly, we observe that not all prompt-based methods effectively enhance a fine-tuned GLIP model. As shown in Tab. 4, methods like MIU-VL and AutoPrompter experience performance degradation when applied to the refined GLIP (MIU-VL: 56.6 → 50.7 AP, AutoPrompter: 56.6 → 53.3 AP). This decline likely occurs because these methods treat prompts as simple contextual information for the target name. During fine-tuning, only the target name is used as the linguistic input, causing a significant distribution shift when prompts are introduced during inference. In contrast, StructuralGLIP encodes prompts into a latent knowledge bank via the auxiliary branch, where prompts are used to construct structural representations during vision-language fusion. However, the final RPN inference still relies on the target name representation. In this way, StructuralGLIP incorporates additional knowledge about the target and alleviates the distribution shifting problem at the same time. This approach allows StructuralGLIP to achieve further performance gains on fine-tuned GLIP (56.6 → 59.3 AP).

## 4.4 Ablation and analysis

**Prompt as Knowledge Bank.** StructuralGLIP uses a dual-branch architecture and mutual selection mechanism to encode prompts into a latent knowledge bank, effectively supporting category-level prompts with rich attribute knowledge. To validate this, we directly feed category-level prompts of StructuralGLIP and instance-level prompt of MIU-VL for a vanilla GLIP to gain its performance with category-level prompt and instance-level prompt, respectively. As shown in Tab. 2, when employing category-level prompt, GLIP suffers a performance degradation (25.9→24.8) while StructuralGLIP gains additional performance improvement (39.2→48.3). This indicates that mutual selection helps StructuralGLIP effectively leverage category prompts by selecting the most relevant information for each image. **Besides**, another important advantage of embedding prompts as a knowledge bank is that this design enables the precise integration of additional knowledge without affecting the distribution of the target representation. To validate this, we conducted ablation studies on the fine-tuned GLIP model, where only the target name is used during the training phase. Then, we evaluate the performance of GLIP and StructuralGLIP under a zero-shot enhancement setting. We present the experimental result

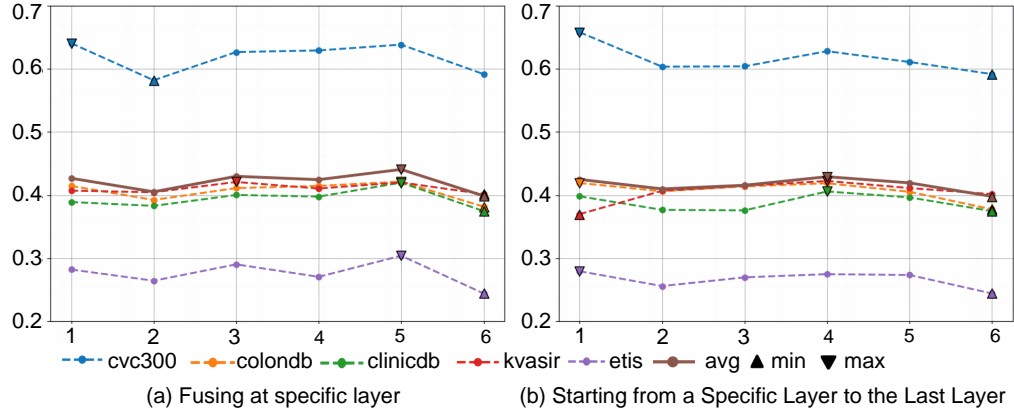

(a) Fusing at specific layer      (b) Starting from a Specific Layer to the Last Layer

Figure 4: Ablation towards the fusing layer of the proposed method

in Tab. 3. Similar to the analysis in Tab. 2, the proposed StructuralGLIP effectively incorporates the knowledge from the bank without a performance degradation. As discussed in Sec. 4.3, with dual-branch architecture, the knowledge bank functions as a residual feature in thee modality fusing phase, which prevents the distribution shift by encoding prompts separately from the target representation, ensuring smooth integration of prompt knowledge during inference.

**Category-prompt generation methods.** To validate the effectiveness of incorporating the prompt generated with the large language model, we exhibit the comparison of only using the VQA model and combing the results of VQA and GPT without fine-tuning the GLIP model in Tab. 4.4. Our experimental results show that GPT can provide more comprehensive knowledge about the target and further improve the performance.

**Ablation on the fusing layer.** We performed an ablation study to analyze how the layer at which the latent knowledge bank is fused impacts the performance. Fig. 4 illustrates two fusion strategies. **The first strategy**, shown in Fig. 4(a), explores the effect of fusing at specific layers. The results indicate that fusion at Layer 5 yields the highest performance across multiple datasets, suggesting that this layer contains the most relevant features for effectively incorporating prompt knowledge. In contrast, earlier layers such as Layer 1 and Layer 2 exhibit lower performance, likely due to their focus on low-level features that are less compatible with the semantic richness of the prompts. **The second strategy**, depicted in Fig. 4(b), investigates the effect of starting from a specific layer and fusing through to the last layer (Layer 6). The results reveal a hierarchical pattern (Fusing Layer4-Layer6>Layer5-Layer6>Layer6), where starting fusion from Layer 4 and continuing to Layer 6 achieves the best results. This indicates a progressively integrating the prompt knowledge at deeper layers allows the model to better utilize the information from the knowledge bank, rather than directly fusing at the last layer. A further analysis of this hierarchy characteristic is conducted in Appendix E, and more insights into the improvement of StructuralGLIP are shown in Appendix H.

**Ablation on $Q$ and $P$.** We explore the joint effects of hyper-parameters of the selected visual tokens and prompt numbers $Q$ and $P$ on CVC-300 under zero-shot detection without the fine-tuned model. As shown in Tab. 4.4, the model achieves the optimal performance with $P = 10$ and $Q = 10$. Further increasing $P$ or $Q$ introduces redundant information and degrades performance.

**More important analysis** towards the category-specific prompt, selection mechanism, the effect of the selected LLM to generate prompts are attached in Appendix A.

## 5 CONCLUSION AND LIMITATION

We introduced StructuralGLIP, a novel zero-shot medical detection model that achieves fine-grained alignment between target descriptions and medical images. Unlike prior works that directly transfer vision-language models to the medical domain, we extended zero-shot medical detection to a more practical setting by exploring both category-level prompts and zero-shot enhancement. Through extensive experiments, we demonstrated that StructuralGLIP excels under these conditions, significantly outperforming existing methods. In future work, we aim to extend the applicability of StructuralGLIP to more diverse medical and non-medical domains, potentially improving its adaptability to varied visual conditions and more complex multimodal tasks.

## 6 ACKNOWLEDGE

The work was supported by the National Key Research and Development Program of China (Grant No. 2023YFC3306401), and the Beijing Natural Science Foundation (No. L244043), and the National Natural Science Foundation of China (No. 62406298); Result in Section 4.4 was supported by the Analytical Center for the Government of the Russian Federation (agreement identifier 000000D730324P540002, grant No 70-2023-001320 dated 27.12.2023).

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

## A   MORE IMPORTANT ANALYSIS AND ABLATION.

**Can category-specific prompts be applied to all methods?** We compare the performance of several methods when category-specific prompts, generated by BLIP and GPT4, are used. Tab. A presents the AP and AP@50 metrics across multiple datasets, including CVC300, ClinicDB, and Kvasir.

Table A: AP and AP@50 Performance comparison of different methods when applied category-level prompt (BLIP + GPT4).

| Method | CVC300 AP & AP@50 | ClinicDB AP & AP@50 | Kvasir AP & AP@50 |
|---|---|---|---|
| StructuralGLIP | 63.9, 89.8 | 42.0, 57.0 | 42.0, 50.5 |
| MIU-VL | 34.3, 53.2 | 17.9, 26.5 | 22.3, 39.4 |
| AutoPrompter | 30.9, 37.4 | 15.8, 31.3 | 17.5, 26.2 |

The results presented in Tab. A clearly demonstrate the superior performance of StructuralGLIP when applied with category-specific prompts, especially in terms of both AP and AP@50 metrics. StructuralGLIP achieves significantly higher detection performance across all datasets compared to MIU-VL and AutoPrompter. For instance, on the CVC300 dataset, StructuralGLIP reaches an AP@50 of 89.8, which is nearly 36% higher than MIU-VL and more than 50% higher than AutoPrompter.

Additionally, StructuralGLIP's performance on the ClinicDB and Kvasir datasets also outperforms the other methods by a substantial margin. These findings suggest that StructuralGLIP is particularly effective in adapting to category-specific prompts and achieving high-quality results across a variety of medical image datasets. On the other hand, both MIU-VL and AutoPrompter show much lower performance, particularly in the AP@50 metric, indicating that their methods do not leverage category-specific prompts as effectively as StructuralGLIP. This underperformance highlights the robustness of StructuralGLIP's approach in utilizing detailed prompts to enhance model accuracy.

In conclusion, these results further reinforce the importance of StructuralGLIP's capability to utilize category-specific prompts effectively, making it the method of choice for high-performance detection tasks in medical imaging.

**Can the selection mechanism help against the noisy knowledge?** In practice, we can not guarantee that the used prompt is precise and clean. Therefore, how well a method can perform on a noisy prompt is very important. To evaluate the robustness of the mutual selection process, we conducted additional experiments on the BCCD dataset, which includes red blood cells, white blood cells, and platelets. In these experiments, we introduced noisy knowledge by mixing attributes from unrelated categories and tested the performance of StructuralGLIP. The results are shown in Tab. B. In this experiment, "(X, X)" denotes using prompts solely for category X during detection of category X, while "(X, Y)" indicates the introduction of prompts from category Y when detecting category X.

Table B: StructuralGLIP's AP@50 performance with noisy knowledge under zero-shot detection setup on BCCD datasets.

| Model | / | Red Blood Cells | White Blood Cells |
|---|---|---|---|
| StructuralGLIP | **Red Blood Cells** | 32.7 | 32.4 |
| | **White Blood Cells** | 60.5 | 61.0 |
| GLIP | **Red Blood Cells** | 21.1 | 15.6 |
| | **White Blood Cells** | 28.3 | 38.7 |

The results show that GLIP suffers significant performance degradation when noisy knowledge is introduced, while StructuralGLIP maintains high accuracy. This demonstrates the robustness of the mutual-selection mechanism, which effectively filters out irrelevant information and selects the most relevant prompts for the task at hand.

**How to assess the quality of the generated prompt?** We evaluate the quality of prompts using CLIP-Score, which measures the cosine similarity between the embeddings of cropped regions (e.g., disease regions) and their corresponding prompts. This evaluation was extended to compare the prompts generated by different multi-modal vision-language models (MLLMs), such as BLIP, LLaVa-7b, and Qwen2-VL-7b, where the MLLMs are used as Vision Question Answering (VQA)

models. The results are summarized in Tab. C & D, comparing both instance-level and category-level prompts.

Table C: CLIP-Score and detection performance under zero-shot enhancement setting on TBX-11k.

| Model | BLIP | LLaVa-7b | Qwen2-VL-7b |
|---|---|---|---|
| Instance-level prompt's CLIP-Score | 0.284 | 0.290 | 0.294 |
| Instance-level AP@50 | 0.780 | 0.783 | 0.792 |
| Category-level prompt's CLIP-Score | 0.264 | 0.262 | 0.288 |
| Category-level AP@50 | 0.782 | 0.765 | 0.790 |

Table D: CLIP-Score and detection performance under zero-shot setting on CVC-300.

| VQA Model | Instance-level CLIP-Score | Instance-level AP@50 |
|---|---|---|
| BLIP | 0.259 | 72.8 |
| LLaVa-7b | 0.272 | 73.9 |
| Qwen2-VL-7b | 0.270 | 74.3 |

The results show a clear positive correlation between CLIP-Score values and detection performance (AP@50). This validates the effectiveness of CLIP-Score as a reliable metric for evaluating the quality of prompts. The analysis underscores the utility of CLIP-Score as an additional evaluation metric for VQA-generated prompts.

**Ablation on the choice of LLMs for Category Prompts.** To assess the impact of different LLMs on performance, we evaluated StructuralGLIP using category-level prompts generated by GPT-4, LLaVa-7b, and Qwen2-VL-7b. The results are summarized in Tab. E, comparing their performance across multiple datasets.

Table E: Ablation on the choice of LLMs when generating category prompts for zero-shot enhancement (AP@50).

| LLM Model | TBX-11k | CVC-300 |
|---|---|---|
| GPT-4 | 79.2 | 96.5 |
| LLaVa-7b | 76.5 | 73.9 |
| Qwen2-VL-7b | 79.0 | 89.0 |

The experimental results indicate that the choice of LLM significantly influences the final performance. In contrast, as shown in Tab. C, the selection of different VQA models has relatively minimal impact on the quality of generated prompts. This is because the LLM-based prompt expansion process primarily relies on the model's internal knowledge and memory of the appearance attributes related to the target lesion, rather than utilizing example images.

Therefore, when using LLMs for prompt expansion, selecting high-performing models is crucial to ensure the generation of reliable prompts. Additionally, for detecting rare diseases, prompts generated through LLM expansion may not be applicable, as these models might lack sufficient domain knowledge of less common conditions. This experiment provides valuable insights into the scenarios where LLM-based prompt expansion can be effectively utilized.

**Ablation on the LLM model to generate prompts.** The results in Tab.C&E shows that the performance of StrucutralGLIP can be largely affected under different choices LLM to generate prompt. Therefore, we use Qwen2-VL-7B as the VQA model, which has the best performance in prompt generating, to generate instance-level prompts and compare the performance of StructuralGLIP with AutoPrompter. The results are shown in Tab. F.

The results in Tab. F demonstrate that StructuralGLIP consistently outperforms AutoPrompter in terms of both AP@50 and AP metrics across all datasets. The results in Tab. G demonstrate that StructuralGLIP consistently outperforms AutoPrompter with different vQA models for instance-level prompt generation.

Table F: AP and AP@50 performance of StructuralGLIP and AutoPrompter with different VQA models for Instance-Level Prompts.

|  | Colondb | Kvasir | Etis | Clinicdb |
| --- | --- | --- | --- | --- |
| AutoPrompter AP@50 | 0.513 | 0.431 | 0.240 | 0.318 |
| AutoPrompter AP | 0.353 | 0.347 | 0.178 | 0.233 |
| StructuralGLIP AP@50 | 0.549 | 0.440 | 0.288 | 0.376 |
| StructuralGLIP AP | 0.373 | 0.359 | 0.193 | 0.291 |

These results confirm that the effectiveness of SturcturalGLIP can be transferred well to different quality of prompts.

Table G: AP and AP@50 Performance of StructuralGLIP and AutoPrompter with different VQA Model for Instance-Level Prompts.

| Method | CVC-300 AP@50 | ClinicDB AP@50 |
| --- | --- | --- |
| StructuralGLIP (BLIP) | 72.8 | 38.2 |
| StructuralGLIP (LLaVa-7B) | 73.9 | 37.9 |
| StructuralGLIP (Qwen2-VL-7B) | 74.3 | 37.6 |
| AutoPrompter (BLIP) | 70.6 | 30.6 |
| AutoPrompter (LLaVa-7B) | 70.9 | 30.9 |
| AutoPrompter (Qwen2-VL-7B) | 75.0 | 31.8 |

# B    COMPARATIVE VISUALIZATION

We illustrate the detection results on the ColonDB and BCCD dataset in Fig. E, where we employ the vanilla GLIP for these prompt-based methods. Intuitively, both AutoPrompter and MIU-VL struggle with either over-detection or missing critical targets. This is likely due to the coarse alignment between vision and target representations, leading to false positives and missed detections. For example, in ColonDB, both methods produce inconsistent bounding boxes, failing to accurately localize the polyp. On the other hand, StructuralGLIPdemonstrates more precise localization with category-level prompts, leading to fewer missed targets and improved confidence scores.

We demonstrate example detection results on the ISIC2016 and TBX11K datasets in Fig. B below, where we employ the fine-tuned GLIP and AutoPrompter for comparison. As shown in Fig. B(a), it is evident that vanilla GLIP and AutoPrompter fail to produce correct classification results for lesion detection. In contrast, our method, benefiting from the category-level prompt, makes corrects classification. For radiographic datasets, our instance method achieves higher confidence scores using the same prompts with AutoPrompter.

# C    DATASET INTRODUCTION

We select four types of medical imaging datasets involving eight benchmarks:

1) Endoscopy datasets for polyp detection: ClinicDB Bernal et al. (2015); Fernández-Esparrach et al. (2016), ColonDB Bernal et al. (2012), Kvasir Jha et al. (2020), ETIS Silva et al. (2014). There are 2,248 images and 2,374 bboxes in total. The complete training and validation images for the entire benchmark are 1160 and 290, respectively. And the number of test set images for CVC-300, CVC-ClinicDB, CVC-ColonDB, Kvasir, and ETIS datasets are 60, 62, 380, 100, and 196 respectively. The primary challenge involves highly variable polyp appearances, obscured views due to mucus and bleeding, and low contrast against surrounding tissues.

2) Microscopy dataset: BCCD shenggan et al. (2018) for blood cell detection (white blood cells, red blood cells, and platelets). The BCCD dataset is designed for blood cell detection tasks, including

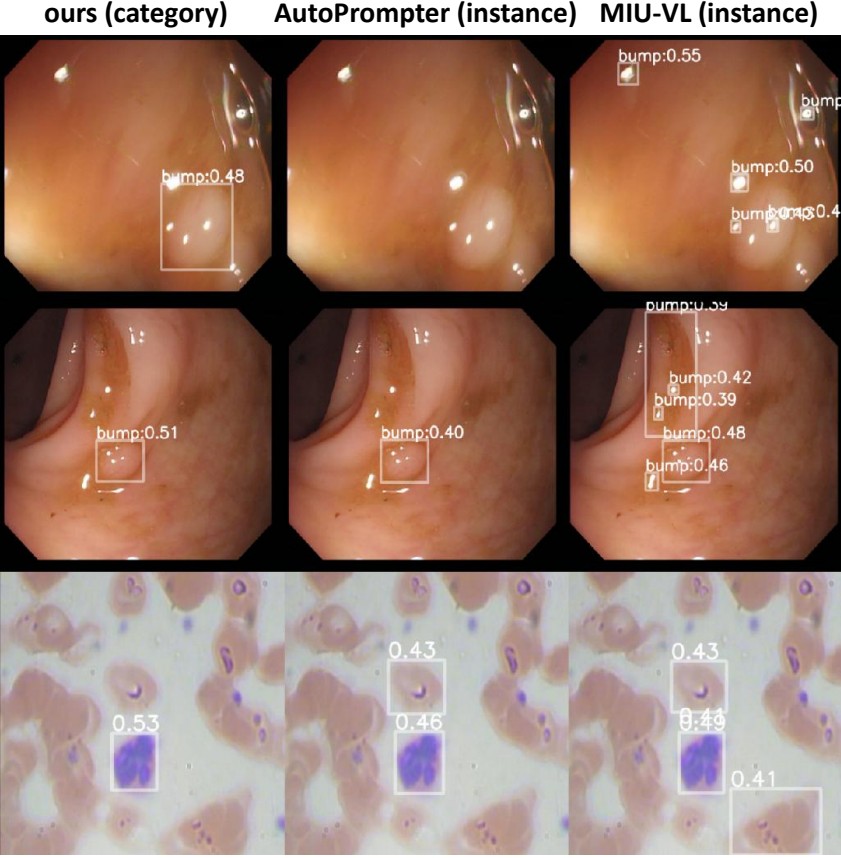

Figure A: Visualization Results on the ColonDB and BCCD datasets.

three classes: white blood cells, red blood cells, and platelets. There are 874 images with 11,789 bboxes for the entire BCCD dataset.

3) Photography dataset: ISIC-2016 for skin lesions detection (benign lesion, malignant lesion). The ISIC-16 dataset consists of 1,279 images with 1,282 bboxes for benign skin lesions and melanoma detection, divided into 720/180/379 images for training, validation, and testing. This dataset pose difficulties due to the small size and high density of the targets, and variations in staining which affect visual clarity and consistency.

4) Radiology image datasets: TBX11k Liu et al. (2023b) for tuberculosis detection in lung x-rays. These datasets are challenging due to the subtle nature of disease indicators, which can obscure key features. The TBX11K dataset is used for tuberculosis detection in the lung, including 799 images and 1,211 bbox labels. Moreover, this dataset is divided into 479/120/200 images for training, validation, and testing sets, respectively.

We demonstrate some example images of these datasets in Fig. F below.

# D  TRAINING DETAILS FOR GLIP'S ZERO-SHOT ENHANCEMENT EXPERIMENT

The zero-shot enhancement aims to further improve the performance of models after supervised training on the downstream datasets. We follow Qin et al. (2022) to use a fine-tuned GLIP model optimized with the Adam optimizerKingma & Ba (2014), where the initial learning rate is set to $1 \times 10^{-4}$ ($1 \times 10^{-5}$ for the BERT text encoder). A weight decay of 0.05 is applied to prevent over-fitting, and the bottom two layers of the image encoder are frozen to preserve fundamental features. Our expressive prompts tailored to the characteristics of the target's appearance are generated using GPT-4 Achiam et al. (2023). Full details of these prompts are available in the Appendix.

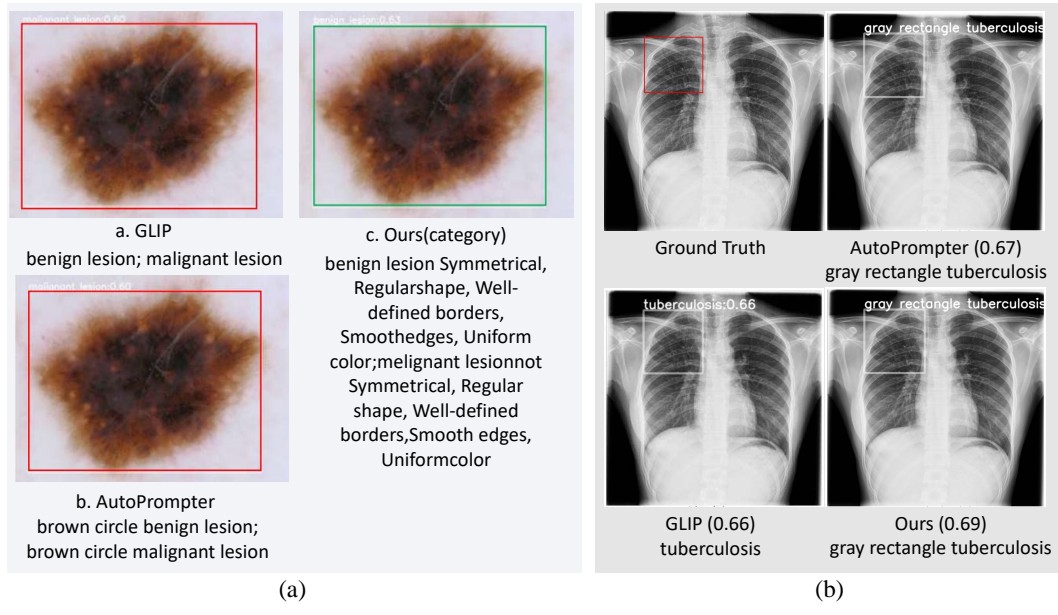

Figure B: Visualization Resultson the ISIC2016 and TBX11K datasets.

# E ANALYSIS FOR THE HIERARCHICAL CHARACTERISTIC OF THE STRUCTURAL REPRESENTATION.

To validate the hierarchical nature of the structural representations derived through layer-wise prompt retrieval, we analyzed the distribution of selected prompts across six GLIP model encoder layers. Specifically, we conducted this analysis using StructuralGLIP with category-level prompts on two medical detection tasks: BCCD red blood cell detection and ClinicDB polyp detection. Prompts were categorized into four types: color, location, shape, and texture (see detailed prompt categories in Appendix G). The results, shown in Fig. D, reveal distinct differences in the frequency of selected prompt types across layers (see Tab. H for the concrete value of the figure). In both tasks, color prompts were consistently selected across all layers, highlighting its importance in medical detection. For ClinicDB, the frequency of shape and texture prompts increased in deeper layers, indicating that these features become more relevant as the model abstracts more complex attributes. In contrast, for red blood cell detection, color remains the predominant feature across layers, while the selection of shape and texture prompts decreases. This analysis demonstrates that StructuralGLIP can dynamically retrieve task-relevant prompts from the knowledge bank at different layers, confirming the adaptable nature of our zero-shot medical detection framework.

| Attribute | colors | | locations | | shapes | | texture | |
|---|---|---|---|---|---|---|---|---|
| Dataset | polyp | red blood cell | polyp | red blood cell | polyp | red blood cell | polyp | red blood cell |
| layer1 | 1436 | 1192 | 183 | 233 | 770 | 474 | 467 | 233 |
| layer2 | 1468 | 1073 | 473 | 17 | 690 | 562 | 572 | 149 |
| layer3 | 1475 | 1109 | 158 | 122 | 702 | 499 | 223 | 136 |
| layer4 | 1505 | 1160 | 436 | 133 | 1131 | 653 | 883 | 245 |
| layer5 | 1498 | 1062 | 416 | 59 | 1163 | 565 | 779 | 269 |
| layer6 | 1450 | 1117 | 182 | 21 | 904 | 433 | 452 | 48 |

Table H: The selected frequencies of different types of prompts across GLIP's different layers

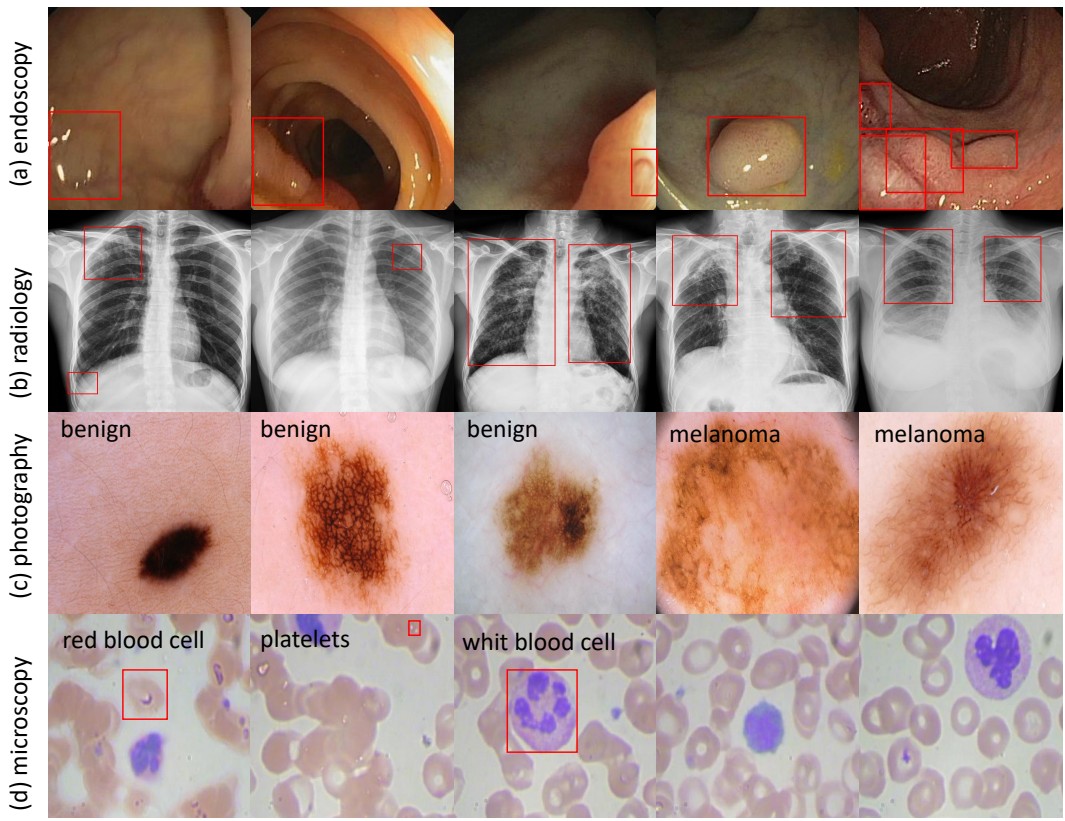

Figure C: Examples of medical images under different imaging conditions.

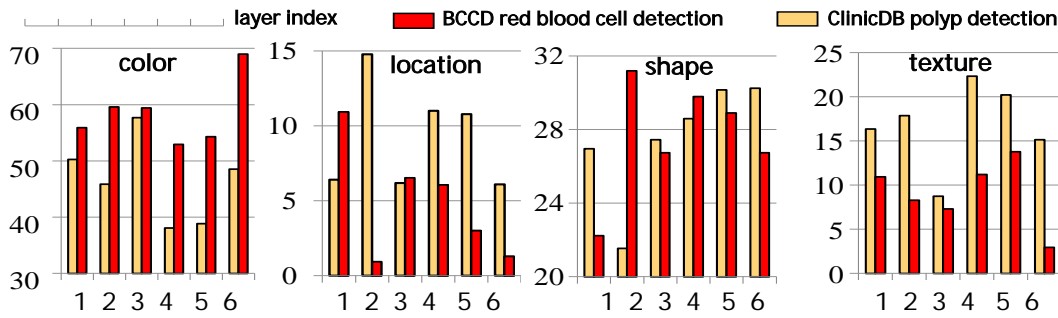

Figure D: Results of the selected frequency (y-axis) of different types of prompts across the network's layer (x-axis).

## F    EFFECT OF PROMPT QUALITY.

As shown in Tab. I, we present the detailed target description (query name with prompt) for red blood cells detection task, and only using the category name "red blood cells" as input to the model of GLIP resulted in poor detection performance, with an AP of just 1.7%. From Tab. I, it is evident that the design of prompt methods significantly affects the ability of vision-language models to utilize prompts for domain enhancement. For instance, our method, with the simple prompt, "pink oval", outperforms MIU-VL, which uses multiple types of attributes, achieving a +7.4%AP50 and +4.3%AP improvement. This improvement is attributed to our structural representation, which achieves hierarchical vision-language alignment, thereby enhancing the utilization of prompts for medical image analysis. Additionally, the structural representation involves fine-grained vision-

language alignment, enabling precise selection of attribute tokens from prompts. This capability allows our method to effectively incorporate more comprehensive prompts, leading to a further improvement of 5.1%AP50 based on the simple short prompt. This demonstrates the effectiveness and robustness of our approach in generating and utilizing prompts for zero-shot detection tasks, showcasing its superiority in achieving efficient and accurate medical image analysis.

Table I: Comparison of different methods with different prompts based on the red blood detection task of BCCD dataset (complete prompts are shown in Appendix G).

| Methods | Target Description (name+prompt) | AP | AP50 |
|---|---|---|---|
| GLIP | [*name*] red blood cells | 1.7 | 4.3 |
| MIU-VL | [*name*] + red color + spherical shape + in birth | 12.0 | 24.7 |
| AutoPrompter | [*name*] + pink oval | 12.6 | 27.0 |
| Ours | [*name*] + pink oval | **16.3** | **34.4** |
| MPT+Cluste | [*name*] + (four prompts below) [flesh-colored, pink, round, blood] | 12.5 | 25.6 |
| Ours (category) | [*name*]+ [*color*] pale bright, *et al.*+ [*shape*] oval round, *et al.* + [*texture*] smooth rough *et al.*+ [*location*] peripheral central, *et al.* | **19.3** | **39.5** |

## G  DETAILED CATEGORY-PROMPT FOR THE MEDICAL DATASETS

| | Colors | Shapes | Textures | Locations |
|---|---|---|---|---|
| Polyp | *white, yellow, orange, red, brown, pink, pale, tan, gray-white, gold, cream, ruby, turquoise, indigo, violet* | *octagon, circle, round, heart, oblong, oval, small, rounded, jagged, wide, large, bulbous, spherical, circular, irregular, diamond* | *smooth, textured, cracked, striped, shiny, dull, speckled, raised, rough, granular, grooved, glossy, veined, pigmented, uneven, mottled, interwoven, lines, patches, complex, reticular, structure* | *rectal, mucosal, elevated, demarcated, creased, folded, isolated, clustered, solitary, honeycombed* |
| Red Blood Cells | *pale, bright red, dark red, pinkish, crimson, ruby, coral, salmon, cherry, scarlet, rusty, maroon, wine, burgundy, rosy, flamingo, peach, copper, mahogany, terracotta* | *disc-shaped, oval, round, elongated, spherical, ring-like, bean-shaped, crescent, irregular, biconcave, elliptical, cuboidal, triangular, squamous, fusiform, polygonal, rod-shaped, fibrillar, amorphous, lobed* | *smooth, rough, granular, fibrous, glossy, matte, sticky, velvety, spongy, creased, crystalline, jelly-like, pitted, wrinkled, spiny, bumpy, flaky, mucous, papillary, striated* | *peripheral, central, upper, lower, medial, lateral, distal, proximal, anterior, posterior, cervical, thoracic, abdominal, pelvic, inguinal, axillary, oral, nasal, occipital parietal* |
| White Blood Cells | *purple, white, pink, gray, blue, translucent, lavender, milky, yellow, pale, clear, light purple, ivory, cream, faint blue, silver, off-white, light gray, opalescent* | *round, oval, irregular, lobed, segmented, spherical, kidney-shaped, amoeboid, polymorphous, triangular, elongated, bean-shaped, cuboidal, crescent, spindle-shaped, fusiform, irregularly-shaped, star-shaped, flattened, discoid* | *granular, rough, smooth, wrinkled, spongy, matte, glossy, fibrous, pitted, veined, speckled, raised, lobulated, ridged, reticular, grooved, folded, striated, flaky, nodular, uneven* | *circulating, peripheral, thoracic, abdominal, pelvic, cervical, axillary, lymphatic, spleen, marrow, mediastinal, proximal, distal, inguinal, occipital, parietal, cranial, vertebral, lumbar, sacral* |

| | | | |
|---|---|---|---|
| Platelets | *yellow, gray, pink, translucent, clear, beige, orange, white, pale yellow, light gray, light pink, golden, amber, straw, ivory, light orange, peach, tan, light brown, opalescent* | *small, round, oval, irregular, disc-shaped, spiked, star-shaped, elongated, granular, fragmented, jagged, ring-shaped, crescent, cuboidal, polygonal, fibrillar, amorphous, fusiform, spherical, irregularly-shaped* | *granular, smooth, rough, spongy, fibrous, pitted, wrinkled, matte, glossy, veined, lobulated, striated, flaky, nodular, reticular, ridged, bumpy, raised, speckled, uneven, lumpy* | *circulating, peripheral, marrow, spleen, liver, thoracic, abdominal, lymphatic, distal, proximal, cervical, axillary, cranial, vertebral, sacral, pelvic, mediastinal, inguinal, parietal, occipital* |
| Benign Lesion | *light brown, tan, pale pink, beige, ivory, light yellow, flesh-colored, clear, translucent, white, pink, light red, off-white, cream, soft yellow, gray, peach, faint brown, faint yellow, light orange* | *round, oval, smooth-edged, well-defined, regular, flat, slightly raised, small, lobulated, dome-shaped, circular, symmetrical, uniform, elongated, flat-topped, irregular, semi-spherical, oblong, disc-shaped, heart-shaped* | *smooth, glossy, matte, uniform, fine, clear, unbroken, even, polished, soft, thin, flat, reticular, striated, nodular, shallow, granular, homogeneous, light-textured, delicate* | *superficial, epidermal, dermal, non-invasive, isolated, peripheral, central, facial, limb, torso, scalp, back, upper, lower, anterior, posterior, lateral, abdominal, neck, arm* |
| Malignant Lesion | *dark brown, black, red, purple, blue, gray, deep red, maroon, dark purple, crimson, burgundy, dark gray, navy, violet, yellowish, pale gray, dark pink, reddish-brown, orange, tan* | *irregular, asymmetric, poorly-defined, multi-lobed, jagged, raised, ulcerated, irregular-edged, large, deep, multi-colored, nodular, star-shaped, rough-edged, uneven, angular, oblong, rough, distorted, fragmented* | *rough, scaly, granular, ulcerated, cracked, irregular, firm, thick, pitted, fibrous, bumpy, crusty, glossy, uneven, speckled, reticular, indurated, papillary, pigmented, veined* | *invasive, dermal, subcutaneous, nodal, systemic, spread, clustered, axial, limb, facial, scalp, back, chest, abdominal, upper, lower, lateral, posterior, anterior, proximal, distal* |
| Tuberculosis | *white, gray, patchy, cloudy, translucent, opaque, pale, faint, bright, shadowed, dull, smoky, hazy, diffused, misty, dense, light gray, speckled, milky, gray-white* | *irregular, nodular, patchy, lobular, diffuse, multi-focal, rounded, asymmetrical, large, small, streaked, segmented, thickened, elongated, fragmented, scattered, spotty, uneven, consolidated, granular* | *rough, fibrotic, granular, nodular, scarred, thick, textured, coarse, uneven, reticular, banded, streaked, fibrous, pitted, grooved, layered, striated, indurated, dense, veined* | *apical, upper lobe, lower lobe, central, peripheral, posterior, anterior, lateral, mediastinal, pleural, diaphragmatic, tracheal, hilum, bronchiolar, thoracic, cervical, upper, lower, rib, clavicle* |

## H ANALYSIS TOWARDS THE FEATURE DISTRIBUTION OF STRUCTURALGLIP.

To provide more insight into the improvement brought by StructuralGLIP, we focus on vision and language features input into the RPN detection model before and after applying our proposed approach. We conduct experiments on five datasets (cvc300, colondb, clinicdb, kvasir, etis) on the vanilla GLIP without fine-tuning and calculate the average value of vision and target representation's attention matrix (termed "average attention strength"). The prompt we used is category-level prompt. Then, we employ a kernel density estimation method to estimate the distribution of average attention strength. We found that there is a significant increase in average attention strength for the proposed StructuralGLIP compared with the vanilla GLIP, indicating better alignment between vision and language representations.

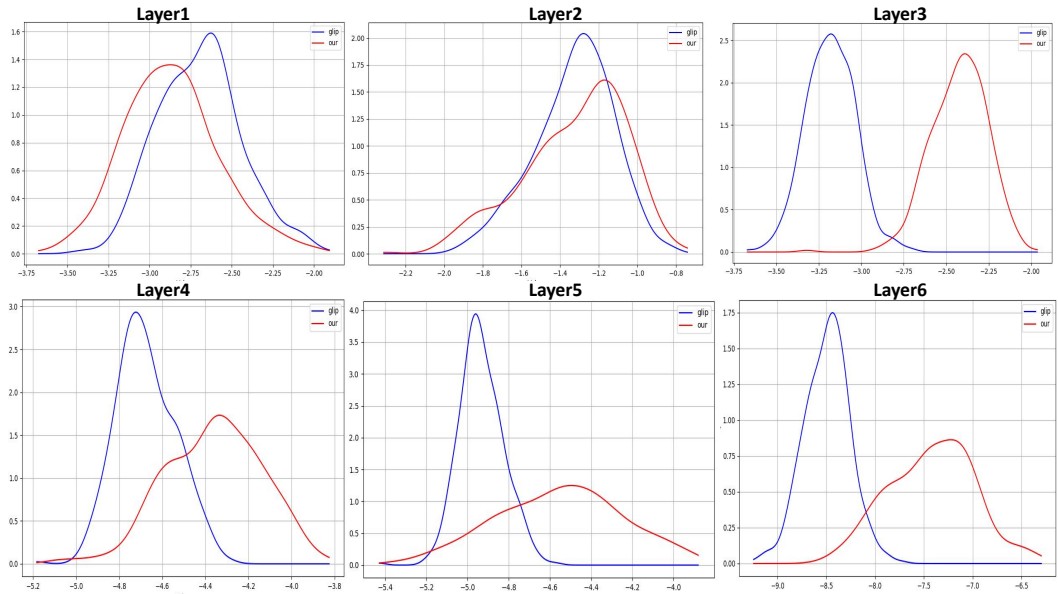

Figure E: Feature distribution of the average attention strength using KDE estimation on five datasets (cvc300, colondb, clinicdb, kvasir etis). The x-axis is the value and the y-axis is the density.

# I  VISUALIZATION ON NATURAL IMAGES

Here we demonstrate the visualization of the proposed StructuralGLIP and GLIP on natural images. The prompt for each category is as follows:

1. Cosmos: broad, delicate, slightly ruffled petals radiating symmetrically around a vibrant yellow center, with a lightweight and airy appearance that contrasts beautifully against the surrounding colors

2. Peony: lush, voluminous, soft, delicate, vibrant, radiant, layered, rounded, ruffled, full, graceful, elegant, eye-catching, rich, luxurious, intricate, symmetrical, silky, and captivating

3. Hematite: flat and irregularly shaped, with a coarse and slightly grainy surface indicative of its iron-rich composition

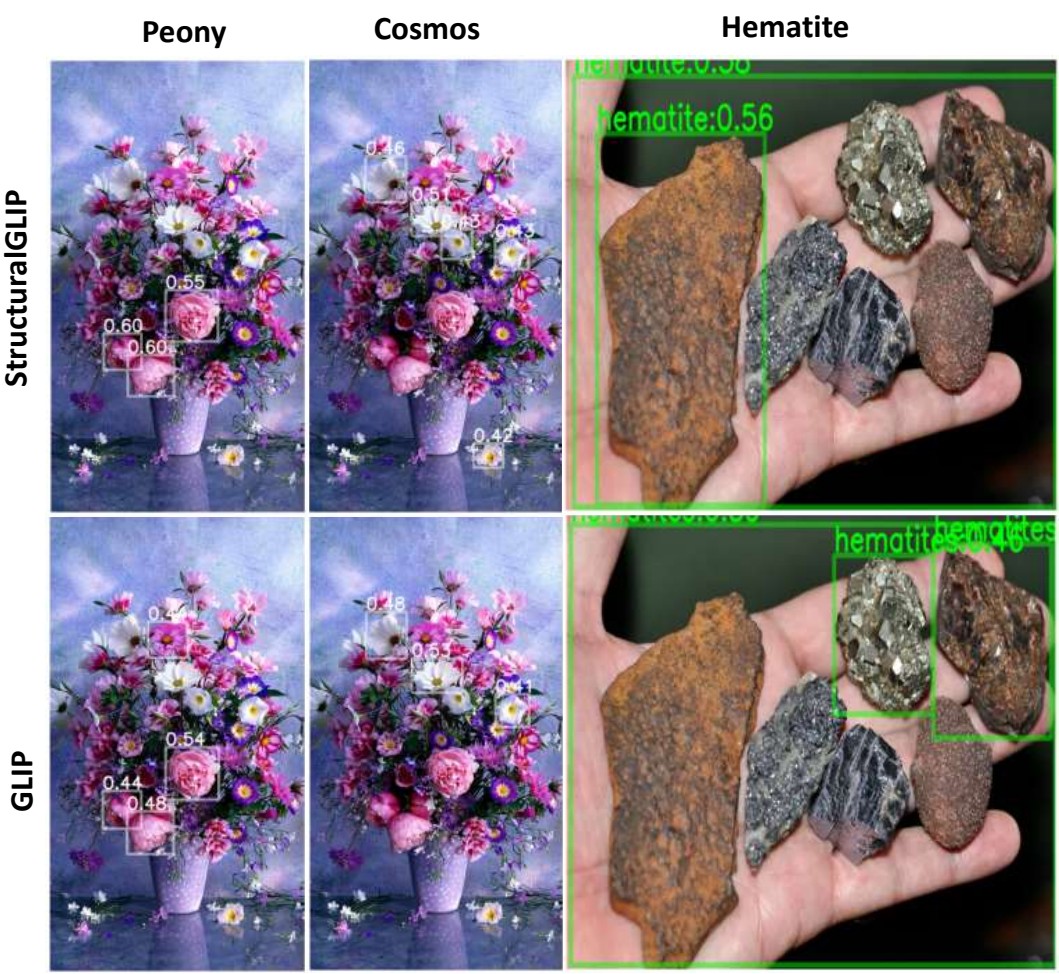

Figure F: Examples of natural images with GLIP and StructuralGLIP.

