# OpenReview forum: "Prompt as Knowledge Bank: Boost Vision-language model via Structural Representation for  zero-shot medical detection"
_ICLR.cc/2025/Conference — ICLR 2025 Poster_

### Official Review · Reviewer_Ky5o · 2024-11-01

**Soundness:** 2
**Presentation:** 3
**Contribution:** 2
**Rating:** 5
**Confidence:** 4

**Summary:**

This paper introduces a zero-shot object detection model, StructrualGLIP, for medical image-based object detection tasks. The method uses a knowledge bank that stores encoded text prompts, which are later used to select and match relevant image features to form fine-grained structural representations to allow better alignment of image features with the prompt information, achieving accurate, context-aware detection.

**Strengths:**

1. the paper demonstrates a comprehensive evaluation of four medical image modalities.
2. the paper's representation and structure are mostly clear and easy to follow (although with some language and word choice issues, which will be discussed below).

**Weaknesses:**

1. The claimed novelty appears to be on the latent knowledge bank and its function to store encoded prompt tokens and later be used at each encoder layer as a vision token selector and vice versa. This is coined as a mutual selection process. The selected information is then merged into the original image and text encoder back at each layer. The entire process is different to a standard contextual prompt method (Fig. 1(a)) but it feels like a quite incremental difference which I don't see much novelty.
2. The word choice of "medical detection" bugs me, in medical science detection refers to "detecting diseases" whereas here this is object detection in medical images so this may cause confusion to certain readers.
3. Line 234 looks unfinished.
4. Eq. 6&7, perhaps the Top-P/Q^{max} function can be simplified by using argmax/argsort function?
5. L201/534: "... fine-grained alignment between target descriptions and medical images", L:256 "forming fine-grained structural representations", can the authors clarify what "fine-grained" refers to in those places?
6. L294: "...like BLIP Li et al. (2022a)", is the VQA model BLIP or not? Have you considered other VQA models and would the performance of other VQA models fluctuate your detection results?
7. L177: I don't entirely agree that zero-shot detection has a real-world clinical need as the clinicians I've encountered would not trust zero-shot settings, in the medical domain, accurate detection/segmentation/diagnosis is the the most important thing.

**Questions:**

Please address Weaknesses #1,5,6.


------------------
04/12/2024:
I thank the authors' detailed responses. Since the reviewer's post deadline has passed so I am writing my final comments here.

My conclusion is that I will retain my original score. I believe the paper should improve clarity as many new arguments and technical details surfaced during the discussion period which should have been included in the main manuscript. The novelty and settings should be explored in more depth.

My comments are:

1. Novelty.

I agree with the authors' argument the image and language encoders do not need to be fine-tuned in your zero-shot detection setting. This I already mentioned in my last comment: "I can understand if the auxiliary branch's language encoder is untrained". My question was whether the modules in Eq. 8 to 10 need training or not? Especially the MHA module in Eq. 8, GLIP has the X-MHA module, did the author reuse the X-MHA weights for the MHA without retraining? The authors mentioned in L263 "we employ a multi-head attention (MHA) mechanism" which reads like it is a new module by the authors' own design. Given an MHA module (as far as I know) should have learnable weights but the authors claim the zero-shot detection setting is training-free, then I find the logic here is contradicted. Furthermore, the authors quoted L210 to demonstrate the training-free paradigm, but that is immediately after Eq.4 and the sentence only describes the encoders. This is an imprecise response to my question so I'm not making a judgment here, but this does make me think the paper should improve clarity.

The above was not the main point of my original novelty concern, but a spin-off. My original point was that if the authors wanted to demonstrate the "preventing/addressing domain shifting" argument for the novelty, you should consider measuring the actual feature distribution difference between GLIP and StructureGLIP.  The authors revealed in the last comments that the domain referred to the features before RPN, and that is what I think you should demonstrate. To recap the authors' comments, the authors claimed the other methods would concatenate prompt so the feature to RPN becomes CLS token of [target_name, prompt] which causes the domain shift, whereas StructualGLIP did the prompts integration in early layers so StructuralGLIP only has CLS of [target_name] to the RPN, and that addresses the domain shift. Conceptually, maybe it is true but please provide empirical evidence. The final performance is a surrogate measure, which does not directly support your domain shift argument. Finally, StructrualGLIP also fuses additional information in the process, so the CLS token should exhibit some domain shift as you are trying to do domain adaptation, these two arguments also have contradictions to each other.

Regarding the "addressing noise prompt" argument, I'd imagine the top-P/Q has a certain capability of limiting noise prompts as you only choose the top tokens, so even when you add more prompts, they won't be selected if the network was already trained to have higher attention for the target classes tokens, but that on the other hand, make it trivial argument rather than a novelty.

7. Zero-shot enhancement. After reading the authors' most recent comments, I strongly feel the setting should not be called zero-shot. When the target dataset's labelled data are already used to fine-tune your GLIP base model, then this is weakly supervised. In L175-176, the authors state "We propose a zero-shot enhancement setting. This involves fine-tuning the model on medical datasets first, and then using prompts to further improve performance on unseen medical images", initially I considered this as the model was trained on relevant datasets such as same/similar imaging modalities but different tasks or different datasets. However, if the target dataset itself was used for finetuning for training, then the "further improve performance on unseen medical images" does not hold.

Other comments:
The experiments I requested were for the understanding of how much performance gain was attributable to the design/novelty of StructrualGLIP, i.e., the introduction of the auxiliary branch. The zero-shot enhancement setting (i.e., fine-tuned GLIP) has boosted the performance by a larger margin than the StructualGLIP can bring to GLIP. Also, the choice of the VQA can sometimes affect the AP/AP50 by a noticeable margin (i.e., that 75 AP) but it appears not stable. Then, at least for me, the viability of zero-shot medical detection unfortunately remains questionable.

Once again I appreciate the authors' effort to address my comments but I'm sorry I will retain my original score based on my thoughts above. The authors are encouraged to delve into the details of the novelty and concepts as well as improve clarity of the manuscript.

---

> ### Author Response · Authors · 2024-11-13
>
> Thanks for your detailed review! We will organize our feedback very soon!!

---

> > ### Author Response · Authors · 2024-11-23
> > **Weakness5**
> >
> > Thank you for raising this important question. The term "fine-grained alignment" in our paper refers to the model's ability to establish correspondence between **local visual patches** in the image and **linguistic tokens** in the prompt bank.
> >
> > In Figure 1(b), we illustrate how the mutual selection mechanism identifies a subset of highly relevant image patch embeddings and linguistic token embeddings from the knowledge bank. These selected tokens and patches are then passed through cross-modal multi-head attention (X-MHA) to inform the main branch's modality fusion process. This approach contrasts with Figure 1(a), where global representations of the image and the knowledge bank are directly fused without considering specific, localized features.
> >
> > By selecting and aligning these local features, our method captures nuanced relationships that may be overlooked in global representations, enabling context-aware feature fusion. This is what we define as "fine-grained alignment."

---

> ### Author Response · Authors · 2024-11-23
> **Weakness1**
>
> Thank you for your thoughtful assessment of our methodology. We appreciate your detailed summary of our approach and the recognition of its differences from standard contextual prompt methods. While we acknowledge that the operations themselves may not appear fundamentally novel, we believe our method introduces **key insights and contributions** that address overlooked challenges in the field, setting it apart from prior work.
>
> **1. Overcoming the domain adaptation limitations of prompt-based methods:**
> Existing prompt-based methods typically input both the target object and prompts as equivalent linguistic contexts into the VLM's language encoder (Fig. 1(a)). However, this approach fundamentally alters the linguistic representation of the target object, making the combined representation a mix of the target and the prompt. This introduces significant challenges during domain fine-tuning:
>
> - During fine-tuning, only the target name is used as linguistic input, resulting in a linguistic representation optimized solely for the target.
> - During inference, the introduction of prompts causes a distribution shift in the linguistic representation, leading to degraded performance.
>
> **As you correctly pointed out in Weakness 7, zero-shot detection methods alone may not be reliable in clinical scenarios, and domain-specific fine-tuning is often essential. Unfortunately, the aforementioned limitations of prompt-based methods significantly hinder their practical application in fine-tuned VLMs.**
>
> StructuralGLIP effectively addresses this challenge by isolating prompt encoding in a separate auxiliary branch. The linguistic knowledge from prompts is incorporated into the main branch via residual connections, allowing the target object representation to remain consistent. This design ensures that our method is compatible with fine-tuned VLMs without introducing distribution shifts. As demonstrated in *Sec. 4.3 (StructuralGLIP facilitates further improvement on fine-tuned models)*, our method achieves significant performance improvements even when applied to fine-tuned models, making it both robust and practical.
>
> **2. Addressing the overlooked impact of noisy prompts:**
>
> Most existing works [1, 2, 3, 4] assume that all generated prompts are beneficial and treat prompts as static, context-expanding knowledge to enhance vision-language model (VLM) performance. However, this assumption neglects the **potential negative impact of noisy or irrelevant prompts**, which can misguide the detection process, especially in medical applications. Our approach is unique in addressing this challenge by:
>
> - Modeling prompts as a latent knowledge bank that decouples prompt encoding from the target representation-
> - Introducing a mutual selection mechanism that dynamically filters and selects only the most relevant prompts for each input.
>
> To validate the robustness of the mutual selection mechanism, we conducted additional experiments on the BCCD dataset, which includes red blood cells, white blood cells, and platelets. We introduced noisy prompts by mixing attributes from unrelated categories and evaluated StructuralGLIP. The results are summarized in Tables S4 . Here (X, X) means we only employ prompts for X when detect X. (X, Y) means we mix the prompts for Y when detection X (X $\neq$ Y).
>
> **Table S4: StructuralGLIP’s AP@50 performance with noisy knowledge under zero-shot detection setup.**
>
> |       | Red Blood Cells | White Blood Cells |
> | ----- | --------------- | ----------------- |
> | Red Blood Cells | **32.7**            | 32.4              |
> | White Blood Cells| 60.5            | **61.0**              |
>
> **Table S4-2: GLIP’s AP@50 performance with noisy knowledge under zero-shot detection setup.**
>
> |       | Red Blood Cells | White Blood Cells |
> | ----- | --------------- | ----------------- |
> | Red   | **21.1**        | 15.6              |
> | White | 28.3            | **38.7**          |
>
> Our findings reveal that StructuralGLIP maintains its performance when noisy knowledge of other category is added to the prompt. This robustness arises from the mutual selection mechanism, which effectively filters out irrelevant information and retrieves only the most relevant prompts for the input image.
>
> Therefore, we believe StructuralGLIP provides novel perspectives and solutions to challenges in prompt-enhanced vision-language models. By addressing issues such as domain adaptation, distribution shifts, and noisy prompts, our work represents a significant step forward in the practical application of prompt-based methods for medical imaging.
>
> ---
>
> ## References
>
> [1] Zero-Shot Nuclei Detection via Visual-Language Pre-trained Models. MICCAI.
>
> [2] Medical image understanding with pretrained vision-language models: A comprehensive study. ICLR, 2023
>
> [3] Multiple prompt fusion for zero-shot lesion detection using vision-language models. MICCAI, 2023
>
> [4] Zero-Shot Nuclei Detection via Visual-Language Pre-trained Models, MICCAI, 2023

---

> > ### Comment · Reviewer_Ky5o · 2024-11-27
> >
> > I appreciate the authors' comprehensive reply, here are my responses:
> >
> > The authors addressed the novelty question with two arguments, which I summarize as 1) the auxiliary branch prevents the domain shift as evidenced by a significant improvement in performance; and 2) addressing noise prompt. These are not direct responses to the novelty but rather the implication/impact.
> >
> > For 1), the domain shift argument is tricky here. While not an expert on the domain shift, in my opinion, a residual connection, i.e., the addition operation, does not guarantee the domain shift will not happen, unless your auxiliary branch's information is not "very useful" hence ignored by the model and not changing the language/image branch's information content much. This I believe, if the authors want to use it as an argument, should demonstrate the domain is not actually shifting. The authors may compute the feature distribution statistics before/after the residual connection, and the distribution of the "domain" does not change significantly. Using the "significant improvement in performance" as evidence is arguably weak as I could counter-argue it is because your categorical level prompts are adding new information which achieves a better performance.
> >
> > Also, I think the so-called "zero-shot enhancement" is probably a newer and fancier term for fine-tuning and solving the same domain adaptation problem. While the authors claim to solve the domain shifting issue, I believe the domain needs to be shifted at least a little bit to adapt to the medical downstream problems. So I'm not entirely sure claiming the method solves domain shift and domain adaptation at the same time is easy to justify.
> >
> > For 2) would I please ask the authors to provide what prompts were selected and used by StructuralGLIP, and are they meaningful selections in human perception? Please also try using a different dataset's prompts and if it is as the author claims, the performance should drop. Without these, a reader could counter-argue that the choice of prompts doesn't matter much at all, which exhibits not so much change in performance for your cross-use of red/whie blood cells' prompts.
> >
> > In general, the auxiliary branch itself is still not really novel, I think the author would agree. The novelty argument now concerns the "exact form" of prompt integration (the residual connection) that actually addresses the claims: domain shift/adaptation + noisy prompt.

---

> ### Author Response · Authors · 2024-11-23
> **Weakness6**
>
> Thank you for this insightful comment. Yes, we use BLIP as our Vision Question Answering (VQA) model, as mentioned in Line 367-371. We appreciate your suggestion to further validate the robustness of our approach by exploring the impact of different VQA models.
>
> To address this, we conducted an ablation study to compare the performance of various VQA models, including BLIP, LLaVa-7b, and Qwen2-VL-7b. These models were used to generate prompts for both instance-level and category-level detection tasks on the CVC-300 dataset. The results are summarized in Table S6:
>
> **Table S5: Ablation on the choice of VQA model with CVC-300 dataset**
>
> |                      | BLIP | LLaVa-7b | Qwen2-VL-7b |
> | -------------------- | ---- | -------- | ---------- |
> | Instance-level AP@50 | 72.8 | 73.9     | 74.3       |
> | Category-level AP@50 | 85.6 | 87.4     | 86.9       |
>
> As shown in Table S6, our method demonstrates robust performance across different VQA models. The results indicate that the selection of the VQA model has a relatively small impact on the final detection performance, confirming the generalizability of our approach.

---

> ### Author Response · Authors · 2024-11-23
> **Weakness2,3,4,7**
>
> (1) Response to weakness2: Thank you for highlighting this potential ambiguity. We agree that "medical detection" may cause confusion, particularly for readers from a medical science background. To clarify, we will replace this term with more precise phrases, such as "disease detection" or "object detection in medical images," in the camera-ready version to ensure clarity and avoid misinterpretation.
>
> (2) Response to weakness3: We apologize for the oversight, which occurred during manuscript preparation. The correct sentence should read: "$B^{i}$ denotes the representation of the prompt bank at the $i^{\text{th}}$ layer." This has been corrected in the revised manuscript.
>
> (3) Response to weakness4: We appreciate your thoughtful suggestion. While the equations in their current form explicitly describe the mutual selection process for clarity, we agree that simplifying them using `argmax` or `argsort` would improve readability. To minimize any potential misunderstanding or confusion among other reviewers during the rebuttal phase, we will retain the current equations for now and implement these updates in the camera-ready version.
>
> (4) Response to weakness7: We fully agree that **pure zero-shot detection** settings (Fig. 2(a) & (b)) have limited practical utility in clinical environments, as clinicians require high levels of accuracy and reliability. However, this setting serves as an important experimental setup to evaluate the effectiveness of our proposed methods in incorporating prompts to enhance zero-shot detection performance.
>
> To address practical clinical needs, we also introduce the **zero-shot enhancement setup** (Fig. 2(c) & (d)), where StructuralGLIP incorporates additional appearance attributes of the target object using prompts after the base VLM model has been fine-tuned on domain-specific medical data. This hybrid approach demonstrates its practical value by achieving significant performance improvements over standard supervised models such as RetinaNet and Faster R-CNN. This setup has been thoroughly discussed in *Sec. 4.3*.

---

> > ### Comment · Reviewer_Ky5o · 2024-11-27
> >
> > I'll put other comments in one to minimize email spam to everyone:
> >
> > weakness 5: thanks for the explanation, please consider revising the paper accordingly.
> > weakness 6: thanks for showing the results for the other VQA models. The use of LLaVa-7b and Qwen2-VL-7b shows better performance than BLIP tells that the performance of StructrualGLIP is dependent on the VQA model. Would other methods, say, the 2nd place AutoPrompter benefit from these VQA models as well? Also, CVC-300 is the only dataset used here, arguably the dataset that gives the best numerical results, I believe one of the datasets with worse numerical results should also be tested to better demonstrate the "relatively small impact". Hence would you please compare your method with AutoPrompter on CVC-300 and BCCD (or at least another low-performance dataset) with both instance/categorical prompts?
> > weakness 7: As I mentioned in the weakness1 reply, is my understanding of "zero-shot enhancement" being equivalent to model fine-tuning correct? what is the difference here?
> >
> > Rest weaknesses, thanks for addressing them, I have no further questions.

---

> ### Author Response · Authors · 2024-11-27
>
> I largely understand your new concerns regarding the comparison and novelty. As for novelty, we think there are still some misunderstanding and we will soon prepare our clarification.  And for VQA comparison, we will test our method and AutoPrompter on ClinicDB and CVC-300 with different VQA model very soon.

---

> ### Author Response · Authors · 2024-11-30
> **The Second Round of Rebuttal for Weakness 1**
>
> Thanks for your reply. It seems there may have been some misunderstandings, and I hope to clarify these aspects.
>
> ### **1. On Novelty and Innovation**
>
> In our last reply, we only acknowledge that our design is based on ``standard structures'' like cross attention, and would like to emphasize core innovation lies in the **fine-grained alignment of visual and textual information through a structural representation (which has been highlighted in our title)**. This is a critical aspect of our approach and is central to its ability:
>
> 1. Robustness to noisy prompts
> 2. Effective prompt enhancement on fine-tuned models.
>
> This fine-grained structural representation is essential because it addresses key challenges that have not been explored in previous prompt-enhanced works, particularly in the context of domain adaptation and prompt noise management. Thus, we believe our approach makes a novel and valuable contribution to the field.
>
> ### **2. Domain Shift and Residual Connection**
>
> There seems to be a misunderstanding. Our discussion of domain shift was meant to explain why  StructuralGLIP is still effective when it is applied to fine-tuned GLIP.  In our experiments, we observed that previous prompt-enhanced models, such as AutoPrompter and MIU-VL, do not consistently improve performance across all benchmarks after fine-tuning (see Table 4). The key reason for this is that during fine-tuning, only the [target-name] is fed into the text encoder to optimize performance. However, during prompt enhancement, both “[target-name, prompt]” are fed into the fine-tuned GLIP’s text encoder. This inconsistency in input format during fine-tuning and prompt-enhanced stages can lead to performance degradation, as the model is exposed to a format it was not trained on. We hope this clarification helps in better understanding our approach.
>
> ### **3. Prompt Selection and Experimental Design**
>
> We would like to clarify the setup of our experiments. We do not cross-use prompts between different categories, as you mentioned. Instead, we mix prompts from different categories to evaluate the impact of prompt noise on model performance. **Specifically, when detecting Red Blood Cells, we use a mixture of prompts for both Red Blood Cells and White Blood Cells as noisy prompt, rather than using White Blood Cells' prompts directly for Red Blood Cells detection.**
>
> This experimental setup is crucial in demonstrating StructuralGLIP's robustness when it is exposed to noisy or mixed prompts. As shown in Table S4 and Table S4-2, the performance of StructuralGLIP remains relatively stable even when noise is introduced, while GLIP suffers a marked performance decline. This highlights the effectiveness of our mutual selection mechanism in filtering out irrelevant prompt information.
>
> **Table S4: StructuralGLIP’s AP@50 Performance with Noisy Knowledge under Zero-Shot Detection Setup. Here (X, X) means we only employ prompts for X when detect X. (X, Y) means we concat prompts for both X and Y when detection X (X ≠ Y).**
>
> |                          | Red Blood Cells | White Blood Cells |
> | ------------------------ | --------------- | ----------------- |
> | Detect Red Blood Cells   | **32.7**        | 32.4              |
> | Detect White Blood Cells | 60.5            | **61.0**          |
>
> **Table S4-2: GLIP’s AP@50 Performance with Noisy Knowledge under Zero-Shot Detection Setup. Here (X, X) means we only employ prompts for X when detect X. (X, Y) means we concat prompts for both X and Y when detection X (X ≠ Y).**
>
> |                          | Red Blood Cells | White Blood Cells |
> | ------------------------ | --------------- | ----------------- |
> | Detect Red Blood Cells   | **21.1**        | 15.6              |
> | Detect White Blood Cells | 28.3            | **38.7**          |
>
> We hope this clears up the misunderstanding and further demonstrates the effectiveness of our approach in managing noisy prompts.

---

> ### Author Response · Authors · 2024-11-30
> **The Second Round of Rebuttal for Weakness 6&7**
>
> ### **Response to Weakness 6**
>
> We have now provided results comparing StructuralGLIP and AutoPrompter across both a high-performance dataset (CVC-300) and a lower-performance dataset (ClinicalDB) using different VQA models (Table S8-1 & Table S8-2). As can be seen,  StructuralGLIP  consistently outperforms  AutoPrompter  in most cases, even on the lower-performance dataset (ClinicalDB).
>
> **Table S8-1: StructuralGLIP's AP@50 Performance with instance-level prompt generated with different VQA models.**
>
> |             | cvc300 | clinicdb |
> | ----------- | ------ | -------- |
> | BLIP        | 72.8   | 38.2     |
> | LLaVa-7B    | 73.9   | 37.9     |
> | Qwen2-VL-7B | 74.3   | 37.6     |
>
> **Table S8-2: AutoPrompter's AP@50 Performance with instance-level prompt generated with different VQA models.**
>
> |             | cvc300 | clinicdb |
> | ----------- | ------ | -------- |
> | BLIP        | 70.6   | 30.6     |
> | LLaVa-7B    | 70.9   | 30.9     |
> | Qwen2-VL-7B | 75.0   | 31.8     |
>
> Regarding your request to compare the **category-level prompt** performance, we must point out that existing baseline methods do not currently support this type of prompt (see Line 389-415). As such, conducting an ablation study for VQA with category-level prompts under these circumstances would lead to an unfair comparison. Instead, we have provided additional evidence in Table S9 to further support our argument.
>
> **Table S9: AP Performance comparison of different methods when applied category-level prompt (BLIP + GPT4).**
>
> |                       Dataset           | CVC300 | ClinicDB | Kvasir |
> | :----------------------------------------- | :----- | :------- | :----- |
> | StructuralGLIP w. category-specific prompt | 63.9   | 42.0     | 42.0   |
> | MIU-VL w. category-specific prompt         | 34.3   | 17.9     | 22.3   |
> | AutoPrompter w. category-specific prompt   | 30.9   | 15.8     | 17.5   |
>
> As shown in Table S9, StructuralGLIP gains an obvious improvement when using category-level prompt. This is due to we embed prompt as knowledge bank with auxiliary bank, and introduce a mutual selection mechanism to select prompts relevant to the current sample during inference. A more detailed explanation for this can be found in Line 52-76, and Line 389-415.
>
> ### **Response to Weakness 7**
>
> As shown in Fig.2 and Line 173, zero-shot enhancement refers to the process in which we apply prompts to a model that has already been fine-tuned on the target task. Additionally, during fine-tuning, only the target name (e.g., the label of the target object or class) is fed into the text encoder of the VLM, without any prompt included.  We hope this clears up any confusion regarding the difference between zero-shot enhancement and model fine-tuning.

---

> > ### Comment · Reviewer_Ky5o · 2024-12-03
> >
> > I appreciate the authors' effort in addressing my questions. Here are my follow-up comments:
> >
> > Weakness 1. Novelty.
> >
> > I read the new response but sorry I still don't find the novelty arguments convincing.
> >
> > First, the robustness to the noisy prompt. The experiment settings of S4 vs S4-2 may support the noise prompt argument but it also can be explained as "GLIP is not fine-tuned" hence the worse performance of GLIP in general / larger performance drop in off-diagonal. To illustrate my point, I don't suppose the StructureGLIP is training-free given the introduction of the auxiliary branch. I can understand if the auxiliary branch's language encoder is untrained, but the MHA, f, and RPN components in Eq. 8-10 should have some training/fine-tuning somehow, otherwise, it probably won't work out of the box given so much addition of features would shift the feature distribution. Please correct me if I'm wrong. If I'm right, what data (input/label) is used to facilitate the training? Did the GLIP in S4/4-2 also get the same training using that? Otherwise, these experiments cannot really justify the noise prompt argument in my opinion.
> >
> > Second, regarding domain shift, I re-read the paper with the domain shift argument, but I still don't find it relevant given there is no evidence of what is shifted in the model's view, is it feature distribution? Figure E shows your activation distribution changed significantly?? Finally, what mechanism in StructureGLIP addressed the domain shift?
> >
> > Last, the authors did not respond to my "Would I please ask the authors to provide what prompts were selected and used by StructuralGLIP, and are they meaningful selections in human perception?" question. Would the authors comment if this is practically difficult to illustrate?
> >
> > Weakness 6.
> >
> > AutoPrompter + Qwen2-VL-7B achieved 75.0 on CVC, which is higher than StructuralGLIP, can the author explain why? My take from Table S8-1/2 is that all methods compared are heavily dependent on the VQA model used.
> >
> > Table S9: Looks fine but why suddenly choose to report AP instead of AP50?
> >
> > Weakness 7.
> >
> > After the authors' explanation, zero-shot enhancement now sounds like weakly supervised object detection. If this is the case, I would suggest the authors not invent new terminologies. As the training settings are not described clearly in the paper, can the authors shed some light regarding the training required for both zero-shot detection/enhancement settings:
> > 1. what do you mean by finetuned on target task for zero-shot enhancement? E.g., in the experiment of the CVC, were the images and the image labels of CVC used for StructureGLIP's fine-tuning?
> > 2. just to reiterate my question in weakness 1, does StructureGLIP need training/finetuning for the zero-shot detection setting, what components are trained and what data is used?

---

> > > ### Author Response · Authors · 2024-12-03
> > >
> > > Considering the tight timeline (approaching the deadline), I will do my best to address the most urgent issues within the next hour.

---

> ### Author Response · Authors · 2024-12-03
> **Third Round Response to Weakness1**
>
> **It appears that the primary misunderstanding has come to light.**
>
> ## **Question1**
>
>  **We would like to clarify that StructuralGLIP is indeed training-free.** Specifically, the language encoder in the auxiliary branch shares its model parameters with the language encoder in the main branch, meaning there is no additional training or fine-tuning involved for the auxiliary branch (see Line 210). We will further emphasize this point for better clarity.
>
> In the zero-shot detection setting (Fig. 2 (a)~(b)), we directly employ the pretrained GLIP to initialize the parameters of the vision encoder and text encoder in the main branch. Thus, in this setting, the auxiliary branch's text encoder also shares the same weights with the pretrained GLIP's text encoder.
>
> In the zero-shot enhancement setting (Fig. 2 (c)~(d)), we employ a GLIP thas has been fine-tuned on the target dataset to initialize the parameters of the vision encoder and text encoder in the main branch. Thus, in this setting, the auxiliary branch's text encoder also shares the same weights as the fine-tuned GLIP's text encoder.
>
> With this clarification, we believe that your subsequent questions regarding Table S4/4-2 and the second question in Weaknss7 have also been addressed.
>
> ## **Question2**
>
> We would like to clarify the details regarding domain shift from the model’s perspective.
>
> **1. StructuralGLIP Construction in the Zero-shot Enhancement Setting**
>
> Our discussion of domain shift was meant to explain why StructuralGLIP is still effective when it is applied to fine-tuned GLIP (zero-shot enhancement setting). To clarify how StructuralGLIP is constructed in the zero-shot enhancement setting: for the CVC-300 dataset, the zero-shot enhancement process begins by fine-tuning the GLIP model using its labeled data (all prompt-based methods will then build **after** the fine-tuning). During fine-tuning, the text encoder of GLIP receives only the target name (i.e., [target name]) as input. After fine-tuning, we build StructuralGLIP by introducing an auxiliary branch with its own text encoder to process the prompt (Except the weights we use are the fine-tuned GLIP now, other setups can be found in the response to Question 1). However, it is important to emphasize that the model parameters for the auxiliary branch’s text encoder are directly shared with those of GLIP's text encoder, and no additional training is required.
>
> **2. What is Shifted in the Model’s View in the Zero-shot Enhancement Setting?**
>
> In the zero-shot enhancement setting, GLIP is first fine-tuned with [target name] as input to the text encoder, after which prompt-based methods are built on the fine-tuned GLIP. During fine-tuning, we feed the CLS token of the text representation of [target name] into the RPN network. Contextual prompt-based methods, however, introduce the prompt by feeding both [target name; prompt] into the GLIP text encoder. This modification causes a **shift in the feature distribution fed into the RPN network** because the input to the RPN now consists of the CLS token from the concatenated sentence [target name; prompt], which alters the textual feature distribution.
>
> **3. What mechanism in StructuralGLIP Can Address the Domain Shift?**
>
> StructuralGLIP addresses this issue through its dual-branch design. Specifically, the main branch’s text encoder only receives the [target name] input, while the auxiliary branch’s text encoder handles the [prompt]. The textual representation from the main branch is then passed to the RPN network, ensuring that the RPN uses the textual features corresponding solely to the target name. The prompt only influences feature fusion in the main branch and is incorporated into the visual encoder. This strategy minimizes the shift in the textual feature distribution at the RPN input compared to methods that directly concatenate [target name; prompt].
>
> **4. Clarification on Figure E**
>
> Regarding Figure E, we would like to clarify that it represents the distribution of average attention values calculated from the attention matrices of different layers in the main branch of StructuralGLIP (see Appendix G). This figure is not related to the distribution of features fed into the RPN network but instead shows the interaction between visual and language representations at the attention level.
>
> ## **Question3**
>
> We have provided it in the next comment.

---

> ### Author Response · Authors · 2024-12-03
> **Illustration of the selected prompt (ClinicDB)**
>
> Actually, we have provided the analysis of the selected prompts of StructuralGLIP in Appendix E. In our previous response, we did not directly address this question because we felt it was more important to first resolve the key misunderstandings. With the most primary misunderstanding coming out, we further provide the prompts selected by StructuralGLIP as the supplement of Appendix E. To make our results more representative, we have included statistics on the words selected by StructuralGLIP at different layers across the ClinicDB and BCCD datasets. Additionally, please note that if a word is preceded by “##,” it indicates that the tokenizer has split the word into tokens.
>
>
> **Table S10-1 The selected prompts of StructuralGLIP with different layers on ClinicalDB datasets. We have categorized the prompts into color, shape, texture, location 4 types.**
>
> | Layer               | Attributes                                                   |
> | :------------------ | :----------------------------------------------------------- |
> | Layer 1 - colors    | orange: 62, brown: 62, red: 62, tan: 62, cream: 62, gold: 62, pink: 61, ruby: 62, pale: 61, indigo: 56, turquoise: 24, white: 23, gray: 7 |
> | Layer 1 - shapes    | wide: 62, round: 59, small: 54, heart: 48, circle: 35, rounded: 20, irregular: 10, oval: 8, jagged: 7 |
> | Layer 1 - texture   | patches: 62, striped: 62, dull: 60, ##kled: 29, lines: 37, spec: 16, shiny: 10, raised: 6, cracked: 1, pigment: 1 |
> | Layer 1 - locations | elevated: 62, folded: 62, ##eased: 55, isolated: 4           |
> | Layer 2 - colors    | pink: 62, orange: 62, brown: 62, gold: 62, tan: 62, -: 61, pale: 62, cream: 61, red: 62, gray: 62, white: 61, ruby: 56, turquoise: 42, indigo: 1 |
> | Layer 2 - shapes    | heart: 34, small: 34, oval: 31, jagged: 3, circle: 6, round: 8, ##long: 2 |
> | Layer 2 - texture   | patches: 61, shiny: 20, raised: 8, dull: 6, lines: 1, striped: 1, pigment: 1, complex: 1 |
> | Layer 2 - locations | folded: 62, elevated: 62, solitary: 60, ##combe: 56, honey: 50, isolated: 58, ##al: 34, clustered: 44, ##eased: 17, ##cated: 20, ##cos: 5, rec: 5 |
> | Layer 3 - colors    | pink: 62, orange: 62, brown: 57, ruby: 62, turquoise: 49, -: 62, red: 46, indigo: 14, gold: 62, cream: 62, pale: 61, tan: 59, white: 60, gray: 55 |
> | Layer 3 - shapes    | oval: 60, heart: 62, small: 51, ##long: 54, rounded: 41, spherical: 2, bulb: 5, ob: 6, circular: 39, round: 57, ##n: 10, circle: 39, oct: 18, wide: 24, jagged: 6, irregular: 4, ##ago: 1 |
> | Layer 3 - texture   | raised: 55, ##kled: 2, striped: 1, shiny: 2, ##d: 1, texture: 1, dull: 2, patches: 1 |
> | Layer 3 - locations | ##eased: 29, folded: 57, solitary: 26, honey: 19, cr: 6, isolated: 10, ##combe: 4, clustered: 6, dem: 1 |
> | Layer 4 - colors    | orange: 62, pink: 62, brown: 61, red: 18, -: 34, turquoise: 12, ruby: 14, cream: 27, white: 19, gold: 30, pale: 22, tan: 12, gray: 1 |
> | Layer 4 - shapes    | small: 54, heart: 55, oval: 46, ##long: 8, wide: 31, jagged: 22, rounded: 27, circle: 1, round: 3, circular: 1 |
> | Layer 4 - texture   | raised: 62, striped: 34, shiny: 43, dull: 45, patches: 18, ##d: 51, ##kled: 51, vein: 36, cracked: 50, texture: 42, ##ed: 2, rough: 4, complex: 4, gran: 4, lines: 1 |
> | Layer 4 - locations | folded: 62, ##eased: 54, isolated: 62, clustered: 36, cr: 23, solitary: 41, dem: 36, elevated: 62, ##ar: 12, ##al: 13, ##cated: 24, honey: 11 |
> | Layer 5 - colors    | orange: 62, pink: 62, -: 59, ruby: 23, turquoise: 25, cream: 20, brown: 1, indigo: 5, gold: 21, white: 18, tan: 11, pale: 11, gray: 17 |
> | Layer 5 - shapes    | heart: 56, round: 50, circle: 44, oval: 46, small: 61, ob: 3, ##ago: 5, ##long: 9, rounded: 48, wide: 29, jagged: 24, irregular: 4, ##n: 4, oct: 1 |
> | Layer 5 - texture   | raised: 62, patches: 38, ##d: 61, complex: 36, ##kled: 39, vein: 21, texture: 20, cracked: 6, lines: 5, ##ven: 5, ##wo: 10, uneven: 11, ##ular: 3, rough: 5, inter: 5, ##ttle: 13, mo: 2, striped: 5, dull: 12, shiny: 3, ##ed: 1 |
> | Layer 5 - locations | folded: 62, clustered: 60, isolated: 45, ##eased: 59, solitary: 62, elevated: 42, honey: 52, ##ar: 1, ##cated: 5, ##combe: 14, dem: 4, cr: 4, ##tal: 2, ##al: 4 |
> | Layer 6 - colors    | orange: 62, pink: 62, indigo: 58, turquoise: 62, ruby: 62, -: 55, cream: 41, white: 24, gold: 38, tan: 35, pale: 34, gray: 13 |
> | Layer 6 - shapes    | ##ago: 53, circle: 58, heart: 55, oct: 50, ##n: 50, round: 55, ob: 19, oval: 33, small: 29, ##long: 11, rounded: 12, jagged: 17, irregular: 2, wide: 7, bulb: 1 |
> | Layer 6 - texture   | raised: 52, shiny: 48, dull: 33, mo: 8, ##d: 7, gran: 12, uneven: 14, ##ttle: 20, smooth: 40, ##wo: 4, rough: 5, lines: 9, ##ven: 8, spec: 5, groove: 2, patches: 1, ##ular: 1, glossy: 1 |
> | Layer 6 - locations | folded: 42, ##eased: 57, solitary: 49, clustered: 13, honey: 10, ##combe: 5, dem: 5, isolated: 1 |

---

> ### Author Response · Authors · 2024-12-03
> **Illustration of the selected prompt (BCCD)**
>
> **Table S10-2 The selected prompts of StructuralGLIP with different layers on BCCD datasets. We have categorized the prompts into color, shape, texture, location 4 types.**
>
> | Layer               | Attributes                                                   |
> | :------------------ | :----------------------------------------------------------- |
> | Layer 1 - colors    | pale: 36, bright: 36, dark: 36, red: 32, scarlet: 36, wine: 36, rosy: 36, ruby: 36, crimson: 36, copper: 36, maroon: 36, rusty: 36, peach: 36, burgundy: 36, salmon: 36, cherry: 36, mahogany: 36, pink: 28, coral: 34, terra: 24, flaming: 11, ##cott: 6, ##o: 7 |
> | Layer 1 - shapes    | round: 36, oval: 36, crescent: 36, elongated: 36, irregular: 34, -: 27, disc: 25, spherical: 6, shaped: 5 |
> | Layer 1 - locations | ##illa: 36, oral: 35, ##al: 35, ing: 27, ##l: 28, cervical: 16, ##ry: 20, abdominal: 13, lateral: 7, ##uin: 6, nasal: 6, posterior: 4 |
> | Layer 2 - colors    | red: 25, pink: 36, ruby: 36, crimson: 36, pale: 36, ##ish: 36, dark: 36, scarlet: 36, peach: 36, wine: 35, cherry: 29, coral: 33, bright: 35, salmon: 15, ##o: 3, burgundy: 10, copper: 20, rosy: 8, maroon: 10 |
> | Layer 2 - shapes    | crescent: 36, round: 36, oval: 36, spherical: 36, shaped: 23, irregular: 36, ring: 36, disc: 35, bean: 35, like: 33, -: 8, lobed: 25, elongated: 30, ##oid: 24, elliptical: 10, ##cave: 10, cub: 4, bi: 1 |
> | Layer 2 - texture   | crystalline: 36, jelly: 35, sticky: 29, wrinkled: 11, ##eased: 5, pitted: 4, like: 5, -: 1, bumpy: 3, ##y: 2, spin: 1 |
> | Layer 2 - locations | medial: 9, upper: 1, oral: 4, lateral: 2, nasal: 1           |
> | Layer 3 - colors    | red: 36, pale: 36, ruby: 36, dark: 36, crimson: 36, pink: 36, bright: 36, peach: 36, ##ish: 36, wine: 36, scarlet: 36, coral: 36, cherry: 31, burgundy: 29, copper: 31, mahogany: 15, ##o: 14, salmon: 23, maroon: 27, rosy: 19, rusty: 3 |
> | Layer 3 - shapes    | oval: 36, round: 36, spherical: 36, crescent: 36, ring: 34, shaped: 29, bean: 34, irregular: 20, disc: 32, like: 31, -: 22, elongated: 9, elliptical: 5, lobed: 3 |
> | Layer 3 - locations | nasal: 33, oral: 22, lateral: 8, medial: 17, ##al: 2, ##illa: 11, ##pit: 5, abdominal: 11, ##vic: 6, ax: 1, cervical: 2, o: 2, ##mal: 1, posterior: 1 |
> | Layer 4 - colors    | pale: 36, red: 36, bright: 36, dark: 36, ruby: 36, coral: 36, crimson: 36, scarlet: 36, pink: 34, cherry: 24, wine: 35, ##ish: 31, maroon: 25, burgundy: 15, peach: 26, salmon: 17, rosy: 11, rusty: 1 |
> | Layer 4 - shapes    | round: 34, spherical: 32, crescent: 35, shaped: 1, irregular: 33, elongated: 27, bean: 34, oval: 26, ring: 30, ##cave: 8, elliptical: 23, lobed: 27, triangular: 21, -: 8, ##gonal: 19, ##form: 15, like: 14, rod: 14, ##mous: 5, fu: 1, ##oid: 1 |
> | Layer 4 - texture   | sticky: 18, matt: 17, smooth: 16, ##eased: 10, pitted: 10, jelly: 10, ##e: 2, rough: 5, crystalline: 8, bumpy: 3, velvet: 4, wrinkled: 3, ##y: 2, spin: 1, fi: 2, ##ong: 1 |
> | Layer 4 - locations | medial: 31, nasal: 18, ##al: 16, lateral: 19, upper: 4, pro: 9, oral: 4, distal: 6, anterior: 9, cervical: 7, abdominal: 4, ##mal: 2, thor: 1, peripheral: 1, central: 1, ##pit: 1 |
> | Layer 5 - colors    | pale: 36, red: 35, bright: 36, dark: 36, ruby: 36, cherry: 35, pink: 36, peach: 34, crimson: 35, ##ish: 32, coral: 35, scarlet: 33, wine: 16, burgundy: 11, salmon: 14, ##o: 10, rosy: 7, maroon: 19, copper: 1 |
> | Layer 5 - shapes    | round: 32, elongated: 27, rod: 27, ##gonal: 34, fi: 30, shaped: 4, spherical: 10, ##bri: 14, ring: 17, bean: 23, triangular: 5, -: 9, crescent: 18, oval: 4, ##form: 11, ##ph: 10, irregular: 11, like: 11, elliptical: 6, amor: 4, ##ous: 4, poly: 7, ##llar: 3, lobed: 1, ##cave: 2, ##mous: 1, fu: 1 |
> | Layer 5 - texture   | ##ong: 35, ##eased: 33, ##y: 8, velvet: 27, sticky: 18, jelly: 11, mu: 14, spin: 14, crystalline: 7, wrinkled: 1, pitted: 4, bumpy: 14, matt: 6, cr: 2, sp: 2, smooth: 4, ##llary: 1, ##cous: 1, pa: 1, ##ak: 5, fi: 1, ##bro: 1 |
> | Layer 5 - locations | medial: 23, lateral: 5, upper: 2, central: 9, peripheral: 12, anterior: 4, ##al: 2, cervical: 1, ##vic: 1 |
> | Layer 6 - colors    | pale: 36, red: 36, ruby: 36, crimson: 36, pink: 36, coral: 36, dark: 36, ##ish: 36, wine: 36, scarlet: 36, bright: 36, peach: 36, salmon: 36, rosy: 34, maroon: 36, cherry: 36, burgundy: 34, copper: 27, rusty: 16, terra: 3, ##cott: 1, ##o: 17, mahogany: 11, flaming: 1 |
> | Layer 6 - shapes    | oval: 36, disc: 36, crescent: 36, round: 36, spherical: 36, -: 14, ring: 36, irregular: 32, bean: 30, like: 27, shaped: 7, ##cave: 11, elongated: 13, cub: 4, ##oid: 1, ##form: 1, rod: 6, fi: 5, ##si: 3, triangular: 3, ##gonal: 2, ##llar: 7, elliptical: 2, amor: 1 |
> | Layer 6 - texture   | ##ong: 18, velvet: 2, ##y: 1, jelly: 3, bumpy: 2, ##eased: 1 |
> | Layer 6 - locations | ##al: 36, lateral: 36, anterior: 27, posterior: 36, inferior: 27, upper: 35, cervical: 36, abdominal: 36, medial: 36, oral: 6, left: 7, distal: 3, rostral: 1, ##ina: 1, right: 3, thorax: 2 |

---

> ### Author Response · Authors · 2024-12-03
> **Third Round Response to Weakness7**
>
> We hope that after reading our previous explanation to Weakness 1, most of your concerns will be addressed. However, to provide a more direct and detailed response, we would like to elaborate further.
>
> ## **Response to Question 1**
>
>  There is still some misunderstanding regarding this setting. All the prompt-based methods discussed in our paper, including StructuralGLIP, are built upon GLIP. In the case of the CVC-300 dataset for zero-shot enhancement experiments, we first fine-tune the GLIP model using the labeled data from the dataset. During fine-tuning, no prompts will be employed. After this fine-tuning, we then built various prompt-based methods for the experiment, as described in Line 173. We will make this point clearer in future revisions. Therefore, the zero-shot enhancement setting means building the prompt methods with a fine-tuned GLIP. This GLIP is fine-tuned on the target dataset with the labeled data without prompt. As the prompt methods may improve the fine-tuned GLIP's performance, we call it "enhancement", and the building of prompt methods does not require further supervised labels, we call it "zero-shot".
>
> ## **Response to Question 2**
> We would like to emphasize that StructuralGLIP is indeed training-free. Specifically, the language encoder in **the auxiliary branch shares its model parameters with the language encoder in the main branch**, meaning there is no additional training or fine-tuning involved for the auxiliary branch (see Line 210). To clarify, in the zero-shot enhancement setting, only the GLIP model is fine-tuned on the target dataset (e.g., CVC-300). After this fine-tuning, we introduce the auxiliary branch with its own text encoder to process the prompt. However, the model parameters for the auxiliary branch’s text encoder are directly shared with those of the main branch's GLIP text encoder, so no additional training or fine-tuning is required for the auxiliary branch.
>
> We hope this clears up the misunderstanding and provides a clearer explanation of our method and training process.

---

> ### Author Response · Authors · 2024-12-03
> **Third Round Response to Weakness 6**
>
> ## **Response to Question1**
>
> We hypothesize that the higher performance of AutoPrompter + Qwen2-VL-7B (75.0 on CVC) may be attributed to the inherent alignment between certain vocabulary and the CVC-300 dataset’s images. Specifically, for the GLIP model, some words may naturally align with the images in the dataset, enabling good performance without the need for StructuralGLIP's mutual selection mechanism. In contrast, the mutual selection structure in StructuralGLIP only selects a subset of visual and textual tokens, which, while helping with denoising, may result in some information loss.
>
> Furthermore, we believe it is normal for such performance variations to occur on individual datasets. The true effectiveness of a method should be assessed based on its performance across a range of datasets. In Table S10, we provide a comparison of the performance of StructuralGLIP and AutoPrompter with Qwen2-VL-7B as the VQA model on additional datasets, which shows a more comprehensive evaluation.
>
> **Table S10: AP and AP@50 performance of StructuralGLIP and AutoPrompter with Qwen2-VL-7B as VQA model.**
> | dataset   |  colondb | kvasir | etis  | clinicdb |
> |----------|----------|--------|-------|----------|
> | AutoPrompter  AP@50    | 0.513  | 0.431   | 0.240  | 0.318 |
> |AutoPrompter AP    | 0.353  | 0.347   | 0.178  | 0.233 |
> | StructuralGLIP AP@50    |0.549  | 0.440   | 0.288  | 0.376 |
> |StructuralGLIP  AP    |  0.373  | 0.359   | 0.193  | 0.291 |
>
>
> ## **Response to Question2**
>
>  You observe that prompt-based methods are inherently dependent on the VQA model. However, this is a standard assumption in research aimed at enhancing the performance of vision-language models in medical domains using prompts [1, 2, 3, 4]. As long as the same prompts are used across different methods, the comparisons remain fair and consistent.
>
> ## **Response to Questino3**
>
>  Finally, regarding your inquiry about why we report AP instead of AP50, we simply chose to present the AP metric without any particular implication or hidden intention. Additionally, Table S9 already addresses why we do not evaluate category prompts with different VQA methods. Here we add the AP@50 performance towards Table S9 as follows:
>
> **Table S9-2: AP and AP@50 Performance comparison of different methods when applied category-level prompt (BLIP + GPT4).**
> |                       Dataset           | CVC300 AP&AP@50 | ClinicDB AP&AP@50 | Kvasir AP&AP@50 |
> | :----------------------------------------- | :----- | :------- | :----- |
> | StructuralGLIP w. category-specific prompt | 63.9,89.8   | 42.0,57.0     | 42.0,50.5   |
> | MIU-VL w. category-specific prompt         | 34.3,53.2  | 17.9,26.5     | 22.3,39.4   |
> | AutoPrompter w. category-specific prompt   | 30.9,37.4   | 15.8,31.3     | 17.5,26.2   |
>
>
> ---
> ## Reference
> [1] Zero-Shot Nuclei Detection via Visual-Language Pre-trained Models. MICCAI.
>
> [2] Medical image understanding with pretrained vision-language models: A comprehensive study. ICLR, 2023
>
> [3] Multiple prompt fusion for zero-shot lesion detection using vision-language models. MICCAI, 2023
>
> [4] Zero-Shot Nuclei Detection via Visual-Language Pre-trained Models, MICCAI, 2023

---

### Official Review · Reviewer_RBUs · 2024-11-03

**Soundness:** 2
**Presentation:** 3
**Contribution:** 2
**Rating:** 6
**Confidence:** 4

**Summary:**

This paper presents Prompt as Knowledge Bank, a method that encodes prompts into an adjustable knowledge bank, enhancing multimodal models’ accuracy in zero shot tasks by dynamically selecting prompts from prior knowledge. This approach achieves robust performance in high-precision fields like medical imaging, even with limited data.

**Strengths:**

1. By introducing a dynamic knowledge bank that selects the most relevant prompts, the model achieves more flexible and accurate vision-language alignment.
2. This method performs exceptionally well in unsupervised and few-shot scenarios, maintaining high detection accuracy even with limited labeled data.
3. The prompt generation module extracts information from prior knowledge of unseen classes, enhancing the model’s adaptability and making it suitable for tasks across various domains.

**Weaknesses:**

1. The introduction of the knowledge bank and prompt generation module increases computational costs, raising demands on hardware resources.
2. The model heavily relies on prompt quality, and low-quality prompts may negatively impact its performance.
3. The impact of the prompt’s LLM on performance is apparent; however, the paper does not analyze how the choice of LLM (e.g., LLaMA, Gemini) affects the results.
4. The complex prompt selection and knowledge bank structure reduce the transparency of the model's decision-making process, posing challenges for clinical applications.
5. There are formatting errors in the paper, such as an unexpected horizontal line near the number 15 in the table 5.

**Questions:**

1. How does the prompt generation module ensure the creation of valuable prompts for unseen classes without introducing distracting information?
2. How is the accuracy of instance-level contextual prompts generated from visual input for VQA ensured, as this seems crucial to the final experimental performance?
3. How does using alternative LLMs affect the experimental results?

---

> ### Author Response · Authors · 2024-11-24
> **Question1**
>
> Thank you for raising this insightful question. Ensuring valuable prompts for unseen classes is a challenge, especially when relying on automated generation methods like VQA and LLMs. We address this potential issue through two key mechanisms in StructuralGLIP:
>
> **(1) Mutual Selection Mechanism:** During inference, StructuralGLIP dynamically filters and retrieves only the most relevant prompt tokens from the knowledge bank. This ensures that noisy or irrelevant prompts do not interfere with detection. The mutual selection mechanism is fundamental to our method’s robustness against distracting information.
>
> **(2)  Experimental Validation:** To evaluate the robustness of this mechanism, we conducted an experiment on the BCCD dataset, where noisy prompts (e.g., unrelated category attributes) were introduced alongside the relevant prompts. The results in Table S4 demonstrate that StructuralGLIP maintains its performance even in the presence of noise on the BCCD dataset. Here (X, X) means we only employ prompts for X when detect X. (X, Y) means we mix the prompts for Y when detection X (X $\neq$ Y).
>
> **Table S4: StructuralGLIP’s AP@50 performance with noisy knowledge under zero-shot detection setup.**
>
> |                   | Red Blood Cells | White Blood Cells |
> | ----------------- | --------------- | ----------------- |
> | Red Blood Cells   | **32.7**        | 32.4              |
> | White Blood Cells | 60.5            | **61.0**          |
>
>
>
>
> **Table S4-2: GLIP’s AP@50 performance with noisy knowledge under zero-shot detection setup.**
>
> |       | Red Blood Cells | White Blood Cells |
> | ----- | --------------- | ----------------- |
> | Red   | **21.1**        | 15.6              |
> | White | 28.3            | **38.7**          |
>
> The results highlight that the mutual selection mechanism effectively filters irrelevant information, ensuring the prompts contribute positively to the detection process.

---

> ### Author Response · Authors · 2024-11-24
> **Question2**
>
> The quality of instance-level prompts is ensured through two complementary approaches:
>
> **(1) Attribute-Specific Prompt Generation:** The VQA model (e.g., BLIP) generates prompts focusing on attributes directly related to the target's visual appearance (e.g., color, shape). This approach minimizes the reliance on domain-specific knowledge, allowing the method to generalize effectively across datasets.
>
> **(2) Quantitative Evaluation with CLIP-Score:** Inspired by prior works in text-to-image generation [1, 2], we evaluate the semantic alignment between cropped disease regions and their corresponding prompts using CLIP-Score. Table S6 and S7 summarize the results, demonstrating a strong correlation between higher CLIP-Score and better detection performance.
>
> **Table S6: CLIP-Score and detection performance under zero-shot enhancement setting on TBX-11k.**
>
> | VQA Model   | Instance-level CLIP-Score | Instance-level AP@50 |
> | ----------- | ------------------------- | -------------------- |
> | BLIP        | 0.284                     | 75.3                 |
> | LLaVa-7b    | 0.290                     | 76.7                 |
> | Qwen2-VL-7b | 0.294                     | 78.2                 |
>
> **Table S7: CLIP-Score and detection performance under zero-shot setting on CVC-300.**
>
> | VQA Model   | Instance-level CLIP-Score | Instance-level AP@50 |
> | ----------- | ------------------------- | -------------------- |
> | BLIP        | 0.259                     | 72.8                 |
> | LLaVa-7b    | 0.272                     | 73.9                 |
> | Qwen2-VL-7b | 0.270                     | 74.3                 |
>
> The strong correlation between CLIP-Score and detection performance demonstrates that high-quality prompts directly contribute to better detection results, thereby validating the effectiveness of the VQA-generated instance-level prompts. From another perspective, the quality of prompts can also be ensured by analyzing the CLIP-Score of prompts generated by different VQA models. A higher CLIP-Score indicates better semantic alignment between the generated prompts and the visual content, serving as a reliable indicator for selecting the most suitable VQA model.
>
> ## References
>
> [1] Li Y, Liu H, Wu Q, et al. Gligen: Open-set grounded text-to-image generation. *Proceedings of the IEEE/CVF Conference on Computer Vision and Pattern Recognition*, 2023.
>
> [2] Hao Y, Chi Z, Dong L, et al. Optimizing prompts for text-to-image generation. *Advances in Neural Information Processing Systems*, 2024.

---

> ### Author Response · Authors · 2024-11-24
> **Question3**
>
> To assess the impact of different LLMs on performance, we evaluated StructuralGLIP using category-level prompts generated by GPT4, LLaVa-7b, and Qwen2-VL-7b. Table S8 compares their performance across multiple datasets.
>
> **Table S8: Ablation on the choice of LLMs when generating category prompts for zero-shot enhancement (AP@50).**
>
> | LLM Model   | TBX-11k | CVC-300 |
> | ----------- | ------- | ------- |
> | GPT4        | 79.2    | 96.5    |
> | LLaVa-7b    | 76.5    | 73.9    |
> | Qwen2-VL-7b | 79.0    | 89.0    |
>
> From the experimental results, it is evident that the choice of LLM has a considerable impact on the final performance. In contrast, as shown in Table S6 and Table S7, the selection of different VQA models has relatively minimal influenceon the quality of generated prompts. This is because the LLM-based prompt expansion process primarily relies on the model's internal knowledge and memory of the appearance attributes related to the target lesion, rather than utilizing example images.
>
> Therefore, when employing LLMs for prompt expansion, it is crucial to select high-performing models to ensure the generation of reliable prompts. Additionally, for detecting rare diseases, the prompts generated through LLM expansion may not be applicable, as the models might lack sufficient domain knowledge for these less common conditions. This experiment provides deeper insights into the scenarios in which LLM-based prompt expansion can be effectively utilized.

---

> ### Author Response · Authors · 2024-11-24
> **Weakness1,2,3,4,5**
>
> (1) Response to weakness1. We acknowledge that our method introduces additional computational costs. However, StructuralGLIP mitigates this by supporting category-level prompts, which are fixed across images of the same class and can be pre-encoded. This significantly reduces runtime costs, avoiding per-image prompt generation during inference. The auxiliary branch’s GPU memory overhead is also minimal (from 6082 MB to 7428 MB with image batch size=2).
>
> (2) Response to weakness2. We believe our responses to Question 1&2 comprehensively address this concern, demonstrating how the mutual selection mechanism ensures prompt utility and how CLIP-Score can validate prompt quality.
>
> (3) Response to weakness3. Please refer to our response to Question 3, which includes an ablation study demonstrating the robustness of StructuralGLIP across different LLMs.
>
> (4) Response to weakness4. We understand the concern about transparency in clinical applications. But actually, StructuralGLIP can explicitly log the selected prompts and their corresponding visual patches during inference. This allows practitioners to trace how prompts are utilized in the decision-making process. Appendix D includes examples of visualized prompt selection, showing which attributes (e.g., "red color" or "irregular shape") were emphasized during detection. We hope this additional analysis addresses your concern about model interpretability.
>
> (5) Response to weakness5. We apologize for the formatting error in Table 5, which resulted from a LaTeX rendering artifact. This has been corrected in the revised manuscript.

---

> > ### Comment · Reviewer_RBUs · 2024-11-27
> >
> > Thanks for your response. While W4 remains challenging and your reply doesn’t fully resolve it, this is good work, and I have updated my score.

---

### Official Review · Reviewer_wdRS · 2024-11-04

**Soundness:** 3
**Presentation:** 3
**Contribution:** 3
**Rating:** 8
**Confidence:** 4

**Summary:**

This paper introduce a new zero-shot method for medical Vision-Language Models(VLMs) for detecting unknown targets. To address the prompt umatched with the variations in the medcial images, the authors propose a StructualGLIP desgin with a main and an auxiliary branch encoders for text input and introduce a mutual selcetion mechanism.
The author explain that the auxiliary branch would work as a knowledge bank where the main branch can extract latent prompt tokens, while the tokens in the knowledge banked are filtered by the mutual selection process.
Overall, the motivation, method, and performance of this work is good enough, but I still need some explaination for some detail, please refer to the weakness and question section. I will consider adjust my rating based on the authors response.

**Strengths:**

1. This work is aiming for a vital issue in medical image understanding field, which is the generalization capability of foundation models with limit data access. The motivation of improving the existing work is clear and strong, which is the lack of object level prompt and fail to capture the various feature of images during prompt desgin.
2. This work present a novel but efficient method, called StructualGLIP, to increase the model's zero-shot/few-shot performance on various datasets. The desgin of StructualGLIP introduce the knowledge bank and mutual selection process to help prompt design process. This method address several shortcomings of current method and is novela and effective.
3. This method largely increased the zero-shot performance on different medical image datasets across different modalities.

**Weaknesses:**

1. Line 234, the sentence seems not finished.
2. One of the major problem of the proposed method is not trainable as the Top-K selection operation is non-differentiable, while previous work is differentiable and thus finetuning would result in better performance. I would suggest include Reparameterization Trick for Gumbel-Softmax to improve your method. Though this work is good enough as a stand-alone method for zero-shot detection. But I see a potential to achieve better performance.

**Questions:**

1. How to evaluate the quality of the generated prompts by VQA method. As some previous work pointed out, the VLM without medical domain adaptation perform poorly on some medical datasets, especially for radiology datasets.
2. For Prompt as Knowledge Bank ablation study seciton, I would like to see an experiment on the whether adding noisy knowledge (for example, knowledge for another target) would sharply downgrade the StructureGLIP performance. This experiment would the the robustness of the mutual-selection process.

---

> ### Author Response · Authors · 2024-11-23
> **Thanks for your feedback**
>
> Thank you for your thoughtful and detailed review of our work. We deeply appreciate your acknowledgment of our paper’s strengths, particularly in addressing key challenges in medical image understanding and introducing StructuralGLIP. Below, we address your questions, suggestions, and noted weaknesses in detail.

---

> ### Author Response · Authors · 2024-11-23
> **Question 1**
>
> You raise an excellent point regarding the quality of prompts generated by VQA, especially on datasets where domain adaptation is lacking (e.g., radiology). To evaluate the quality of generated prompts, we have devised two complementary approaches:
>
> **(1) Fine-tuning VLM on radiology datasets for evaluation**
> While the original GLIP model performs poorly on radiology datasets like TBX-11k due to a lack of adaptation, we conducted additional experiments using a fine-tuned GLIP model to better evaluate the prompt quality. Table S1 shows the results of using instance-level and category-level prompts under the zero-shot enhancement setting (see Figure 2 (c)-(d)). Notably, the category-level prompts, which are more comprehensive, consistently improved performance over instance-level prompts, validating their quality.
>
> **Table S1: Zero-shot enhancement results on radiology datasets.**
>
> |       | Faster RCNN | RetinaNet | GLIP | MIU-VL | AutoPrompter | Ours (instance) | Ours (category) |
> | ----- | ----------- | --------- | ---- | ------ | ------------ | --------------- | --------------- |
> | AP    | 33.9        | 37.0      | 35.2 | 35.5   | 35.7         | 37.3            | 37.8            |
> | AP@50 | 73.9        | 77.9      | 75.3 | 76.7   | 76.5         | 78.2            | 79.2            |
>
> **(2) Quantitative evaluation using CLIP-Score**
> Inspired by text-to-image generation works [1, 2], we evaluated prompt quality using CLIP-Score, which measures the cosine similarity between the embeddings of cropped regions (e.g., disease regions) and their corresponding prompts. We extended this analysis to compare the prompts generated by different multi-modal vision-language models (MLLM), e.g., BLIP, LLaVa-7b, Qwen2-VL-7b, where the MLLMs are used as VQA model.  Table S3 summarizes the results, comparing both instance-level prompts and category-level prompts.
>
> **Table S3: CLIP-Score and detection performance under zero-shot enhancement setting on TBX-11k.**
>
> |                                    | BLIP  | LLaVa-7b | Qwen2-VL-7b |
> | ---------------------------------- | ----- | -------- | ----------- |
> | Instance-level prompt's CLIP-Score | 0.284 | 0.290    | 0.294       |
> | Instance-level AP@50               | 0.780 | 0.783    | 0.792       |
> | Category-level prompt's CLIP-Score | 0.264 | 0.262    | 0.288       |
> | Category-level AP@50               | 0.782 | 0.765    | 0.790       |
>
> There is a clear positive correlation between CLIP-Score values and the detection performance (AP@50), validating the effectiveness of CLIP-Score as a metric for evaluating prompt quality. This analysis reinforces the utility of CLIP-Score as an additional evaluation metric for VQA-generated prompts.
>
> ---
>
> ## Reference
>
> [1] Li Y, Liu H, Wu Q, et al. Gligen: Open-set grounded text-to-image generation. *Proceedings of the IEEE/CVF Conference on Computer Vision and Pattern Recognition*, 2023.
>
> [2] Hao Y, Chi Z, Dong L, et al. Optimizing prompts for text-to-image generation. *Advances in Neural Information Processing Systems*, 2024.

---

> ### Author Response · Authors · 2024-11-23
> **Question2**
>
> Thank you for this insightful suggestion. To test the robustness of the mutual selection mechanism, we conducted additional experiments on the BCCD dataset, which includes red blood cells, white blood cells, and platelets. For this test, we added noisy prompts by mixing attributes of unrelated categories and evaluated StructuralGLIP. Results are displayed in Table S4. Here (X, X) means we only employ prompts for X when detect X. (X, Y) means we mix the prompts for Y when detection X (X $\neq$ Y).
>
> **Table S4: StructuralGLIP’s AP@50 performance with noisy knowledge under zero-shot detection setup.**
>
> |       | Red Blood Cells | White Blood Cells |
> | ----- | --------------- | ----------------- |
> | Red   | **32.7**        | 32.4              |
> | White | 60.5            | **61.0**          |
>
>
>
>
> **Table S4-2: GLIP’s AP@50 performance with noisy knowledge under zero-shot detection setup.**
>
> |       | Red Blood Cells | White Blood Cells |
> | ----- | --------------- | ----------------- |
> | Red   | **21.1**        | 15.6              |
> | White | 28.3            | **38.7**          |
>
> Our findings reveal that GLIP suffers significant performance degradation when noisy knowledge is introduced, while StructuralGLIP maintains high accuracy. This robustness stems from the mutual selection mechanism, which effectively filters irrelevant information and retrieves the most relevant prompts for the given image.

---

> > ### Comment · Reviewer_wdRS · 2024-11-25
> >
> > Table S4 is the result for StructuralGLIP or the original GLIP?

---

> > > ### Author Response · Authors · 2024-11-25
> > >
> > > Table S4 is the result for StructuralGLIP.

---

> > > ### Author Response · Authors · 2024-11-25
> > >
> > > Table S4 is the result for StructuralGLIP.
> > >
> > > **Table S4: StructuralGLIP’s AP@50 performance with noisy knowledge under zero-shot detection setup.**
> > >
> > > |       | Red Blood Cells | White Blood Cells |
> > > | ----- | --------------- | ----------------- |
> > > | Red   | **32.7**        | 32.4              |
> > > | White | 60.5            | **61.0**          |
> > >
> > > and here we add the result of the original GLIP in the same setting.
> > >
> > >
> > >
> > > **Table S4-2: GLIP’s AP@50 performance with noisy knowledge under zero-shot detection setup.**
> > >
> > > |       | Red Blood Cells | White Blood Cells |
> > > | ----- | --------------- | ----------------- |
> > > | Red   | **21.1**        | 15.6              |
> > > | White | 28.3            | **38.7**          |
> > >
> > >
> > > We can see that original GLIP suffers a significant performance drop when the noisy knowledge is added.

---

> > > > ### Comment · Reviewer_wdRS · 2024-11-27
> > > >
> > > > I have no furhter question, and thank you for your great work. I have raised my scores.

---

> > > > > ### Author Response · Authors · 2024-11-27
> > > > >
> > > > > Thanks a lot! We really acknowledge your thoughtful response in time.

---

> ### Author Response · Authors · 2024-11-23
> **Weakness1,2**
>
> (1) Response to weakness 1: Thank you for pointing this out. The incomplete sentence on Line 234 was a formatting issue during manuscript preparation. The corrected sentence reads: "$B^{i}$ denotes the representation of the prompt bank at the $i^{\text{th}}$ layer."
>
>
> (2) Response to weakness 2: We appreciate your suggestion to incorporate a reparameterization trick like Gumbel-Softmax to make the Top-K selection process differentiable. While our current focus is on leveraging zero-shot methods without fine-tuning, this enhancement could improve the fine-tuning capability of our model, opening new avenues for research. We plan to explore this direction in future work.
>
> However, it is important to note that the non-differentiable selection mechanism does not hinder our model’s performance in zero-shot scenarios. StructuralGLIP achieves state-of-the-art results across multiple datasets, showcasing its robustness and effectiveness even without fine-tuning.

---

### Official Review · Reviewer_cerF · 2024-11-04

**Soundness:** 3
**Presentation:** 2
**Contribution:** 3
**Rating:** 6
**Confidence:** 4

**Summary:**

In this study, the authors introduce StructuralGLIP, a novel approach to zero-shot medical detection using vision-language models (VLMs). This method leverages structured representations within a dual-branch architecture that enables nuanced alignment between images and textual prompts, significantly enhancing the model's adaptability to new medical scenarios without needing annotated data. StructuralGLIP uses category-level prompts, maintained in a latent knowledge bank, and a mutual selection mechanism for precise cross-modal fusion, thus improving accuracy across diverse medical imaging datasets.

**Strengths:**

(1)	The paper introduces an effective structural representation by encoding prompts into a knowledge bank and utilizing a dual-branch structure. This approach enables adaptive and context-aware alignment, which is particularly advantageous for complex medical detection tasks.
(2)	StructuralGLIP outperforms traditional zero-shot models by effectively handling both instance-level and category-level prompts, achieving significant improvements across various benchmarks in endoscopy, microscopy, radiology, and more.
(3)	By allowing for zero-shot enhancement, the model can be fine-tuned and then further improved with category-level prompts, a feature well-suited for dynamic medical settings where data annotation is scarce.

**Weaknesses:**

(1)	The proposed dual-branch structure with a knowledge bank requires complex engineering and computational resources, potentially limiting its accessibility for practitioners in less resource-rich environments.
(2)	The paper may not adequately address the potential data imbalance present in the datasets used for evaluation. Some diseases or conditions may have significantly fewer examples, which could impact the model's performance and generalizability.
(3)	The model's inner workings, particularly regarding how it selects and utilizes prompts, may be difficult for practitioners to interpret, limiting trust in its decisions and making it harder to diagnose potential failures.
(4)	Despite improvements in alignment, there may still be instances of misalignment between visual features and prompts, especially in cases of atypical presentations, which could lead to missed detections.

**Questions:**

(1)	To what extent can the findings be generalized to other medical imaging modalities or less common diseases? Are there plans to evaluate the model on broader datasets?
(2)	Besides Average Precision, what other metrics were considered for evaluating model performance? Are there plans to incorporate user feedback or clinical outcomes in future evaluations?
(3)	This paper focuses on zero-shot medical detection, whereas GLIP was initially developed for natural images. Can the proposed method also be applied effectively to natural image datasets?

---

> ### Author Response · Authors · 2024-11-13
>
> Thanks for your careful review! We will try our best to reply to your concern regarding our method soon!

---

> ### Author Response · Authors · 2024-11-23
> **Question1**
>
> We appreciate this insightful question. The generalization of our method can be discussed from two perspectives:
>
> **Generalization to other medical imaging modalities:**
>
> While this work primarily focuses on endoscopy and microscopy datasets, we indeed plan to extend our method to other modalities, such as CT scans and radiology images. However, as GLIP was originally designed for natural RGB images, our preliminary experiments show limited zero-shot performance on these modalities without further adaptation. To address this limitation, we plan to explore two solutions: (a) using vision-language models pre-trained on modality-specific data (e.g., chest X-rays or CT), or (b) fine-tuning our method on medical imaging datasets.
> For example, Table S1 below shows the effectiveness of our method when applied to radiology (chest X-ray) data after fine-tuning in a zero-shot enhancement setting (Fig. 2 (c)-(d)). All comparisons were conducted with consistent prompts.
>
> **Table S1: Zero-shot enhancement results on radiology datasets.**
>
> |       | Faster RCNN | RetinaNet | GLIP | MIU-VL | AutoPrompter | Ours (instance) | Ours (category) |
> | ----- | ----------- | --------- | ---- | ------ | ------------ | --------------- | --------------- |
> | AP    | 33.9        | 37.0      | 35.2 | 35.5   | 35.7         | 37.3            | 37.8            |
> | AP@50 | 73.9        | 77.9      | 75.3 | 76.7   | 76.5         | 78.2            | 79.2            |
>
> **Generalization to less common diseases:**
>
>
> Our instance-level zero-shot setup (Fig. 2 (a)) is specifically designed to address situations with limited training data. From an evaluation perspective, this setup inherently assesses performance in data-scarce scenarios. From a methodological perspective, our prompt generation pipeline uses VQA and LLMs to generate descriptive prompts covering appearance-related attributes such as color, shape, and texture (Appendix F). However, we acknowledge that for extremely rare diseases, LLMs may not always provide accurate descriptions. In such cases, the prompts can be generated solely using VQA models, bypassing LLMs. The instance-level comparisons in Table 1 already evaluate our method under such conditions, demonstrating its applicability to less common diseases.

---

> ### Author Response · Authors · 2024-11-23
> **Question2&3**
>
> **Question2**: Thank you for raising this point. In addition to quantitative metrics such as AP and AP@50, we conducted a qualitative user study to better understand the practical value of our method. Specifically, we randomly selected 18 test images from the  ColonDB dataset and generated detection results using MIU-VL, AutoPrompter, and StructuralGLIP. A group of participants (N=49) was tasked with ranking the methods based on the perceived accuracy of detection boxes. Participants evaluated factors such as precise localization of the target and avoidance of false positives (e.g., misidentifying shadows or irrelevant regions). The results are summarized in Table S2:
>
> **Table S2: Human evaluation results.**
>
> |           | MIU-VL | AutoPrompter | Ours (instance) |
> | --------- | ------ | ------------ | --------------- |
> | Avg. Rank | 2.1    | 2.6          | 1.3             |
>
>
> The feedback from participants confirms that our method not only achieves fewer false positives but also outputs higher-confidence predictions, leading to a clear preference for StructuralGLIP over baseline methods.
>
> For future work, we plan to incorporate clinical feedback and consider additional metrics, such as clinical usability assessments, to evaluate the model’s real-world applicability.
>
> **Question3**: Yes, our proposed method can also be effectively applied to natural image datasets. We would like to clarify why we chose to focus on the medical domain for prompt enhancement. Our primary motivation was to demonstrate how vision-language models can leverage linguistic information to provide additional object priors, thereby improving detection performance. The medical domain, with its relatively simpler and more focused detection tasks, serves as an ideal environment for validating this idea.
>
> For natural images, given that GLIP has already undergone extensive pre-training, we deliberately selected less common natural image datasets—such as Hematite, Cosmos, and Peony—to assess the generalization capabilities of our method. We have included visualizations in Appendix H to demonstrate how our method performs on natural images.

---

> ### Author Response · Authors · 2024-11-23
> **Response to Weakness**
>
> **(1) Computational complexity.** We acknowledge that introducing an auxiliary branch adds architectural complexity. However, this does not always result in increased computational overhead. First, the GPU memory usage increases due to the auxiliary branch is marginal, from 6082 MB (GLIP) to 7428 MB (StructuralGLIP) with image batch size=2, allowing the model to run efficiently on a standard NVIDIA 3060 GPU. Moreover, our method supports category-level prompts, which are fixed for all images of the same target. This eliminates the need to generate instance-specific prompts for each image during inference, thereby reducing overall computational costs and increasing inference speed (Table 2 (b) & (d)).
>
> **(2) Data imbalance.** This concern has been addressed in our response to **Question 1**. Our method inherently handles data-scarce scenarios through its instance-level zero-shot setup and flexible prompt generation pipeline.
>
> **(3) Interpretability.** We understand your concerns about interpretability. However, we believe StructuralGLIP enhances interpretability compared to prior methods. By decoupling prompt encoding into the auxiliary branch, the selection mechanism makes the utilized prompt words visible to users. This transparency allows practitioners to analyze which prompts were used during inference. For example, in Appendix D (Figure D and Table A), we analyze the selected prompts across different tasks, revealing the attributes the model prioritizes during detection. This provides insights into the model’s reasoning process.
>
> **(4) Misalignment in atypical cases.** We acknowledge that misalignment may occur in atypical presentations. While this limitation is partially mitigated by fine-tuning on specific datasets, as shown in Table S1, we plan to explore anomaly detection techniques and adaptive learning strategies to better handle rare or atypical cases in future work.

---

### Meta-Review · Area_Chair_gvic · 2024-12-20

[review text omitted: it was posted to a different submission]

---

### Decision · Program_Chairs · 2025-01-22

Accept (Poster)